# DDF-HO: Hand-Held Object Reconstruction via Conditional Directed Distance Field

**Chenyangguang Zhang**[1*], **Yan Di**[2*], **Ruida Zhang**[1*], **Guangyao Zhai**[2],
**Fabian Manhardt**[3], **Federico Tombari**[2,3], **Xiangyang Ji**[1]
[1]Tsinghua University, [2]Technical University of Munich, [3] Google,
{zcyg22@mails., zhangrd21@mails., xyji@}tsinghua.edu.cn, {yan.di@}tum.de

## Abstract

Reconstructing hand-held objects from a single RGB image is an important and challenging problem. Existing works utilizing Signed Distance Fields (SDF) reveal limitations in comprehensively capturing the complex hand-object interactions, since SDF is only reliable within the proximity of the target, and hence, infeasible to simultaneously encode local hand and object cues. To address this issue, we propose DDF-HO, a novel approach leveraging Directed Distance Field (DDF) as the shape representation. Unlike SDF, DDF maps a ray in 3D space, consisting of an origin and a direction, to corresponding DDF values, including a binary visibility signal determining whether the ray intersects the objects and a distance value measuring the distance from origin to target in the given direction. We randomly sample multiple rays and collect local to global geometric features for them by introducing a novel 2D ray-based feature aggregation scheme and a 3D intersection-aware hand pose embedding, combining 2D-3D features to model hand-object interactions. Extensive experiments on synthetic and real-world datasets demonstrate that DDF-HO consistently outperforms all baseline methods by a large margin, especially under Chamfer Distance, with about $80\%$ leap forward. Codes are available at `https://github.com/ZhangCYG/DDFHO`.

## 1   Introduction

Hand-held object reconstruction refers to creating a 3D model for the object grasped by the hand. It is an essential and versatile technique with many practical applications, *e.g.* robotics [78, 66, 39, 73, 72], augmented and virtual reality [37], medical imaging [50]. Hence, in recent years, significant research efforts have been directed towards the domain of reconstructing high-quality shapes of hand-held objects, without relying on object templates or depth information. Despite the progress made, most existing methods rely on the use of **S**igned **D**istance **F**ields (SDF) as the primary shape representation, which brings about two core challenges in hand-held object reconstruction due to the inherent characteristics of SDF.

**First**, SDF is an undirected function in 3D space. Consequently, roughly determining the nearest point on the target object to a sampled point in the absence of object shape knowledge is infeasible. This limitation poses a significant challenge for single image hand-held object reconstruction as it is difficult to extract the necessary features to represent both the sampled point and its nearest neighbor on the object surface. Previous methods [67] have attempted to address this challenge by aggregating features within a local patch centered around the projection of the point, as shown in Fig. 1 (S-2). However, this approach is unreliable when the sampled point is far from the object surface since the local patch may not include the information of its nearest point. Therefore, for hand-held object

---

*Authors with equal contributions.

37th Conference on Neural Information Processing Systems (NeurIPS 2023).

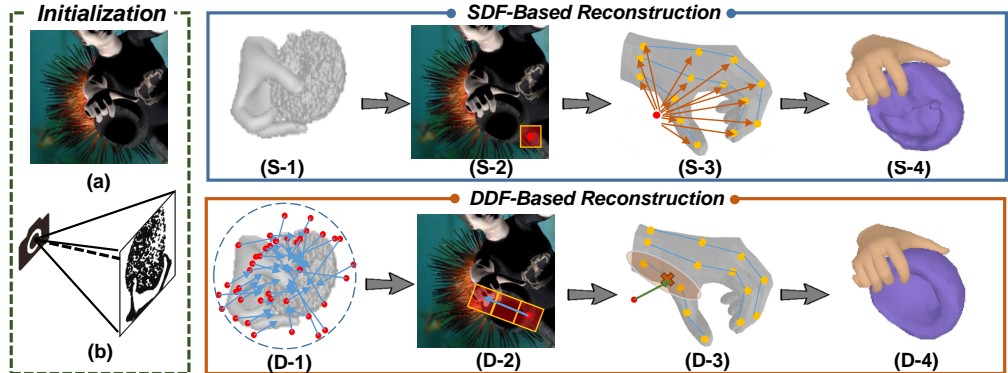

Figure 1: **SDF *vs* DDF based hand-held object reconstruction.** Given an input RGB image (a) and estimated hand and camera pose (b), SDF-based and DDF-based reconstruction pipelines vary from sampling spaces (S-1, D-1) and feature aggregation techniques (S-2, D-2 and S-3, D-3). SDF points sampling space must stay close to the object surface (S-1) or would lead to degraded network prediction results [47], while DDF ray sampling space (D-1) can be large enough to encapsulate the hand and object meshes. SDF methods typically aggregate features for the sampled point $P$ in its local patch, which is not reliable when $P$ is far from the object surface (S-2). DDF, however, aggregates features along the projection line $r'$ for ray $R$, which naturally captures both the information of the point and its intersection with the object surface (D-2). SDF methods cannot directly yield the contact points on the hand surface, so that only global relative hand joints encoding is used (S-3). On the contrast, DDF can get the intersection region of sampled rays and hand surface, leading to more representative local intersection-aware hand encoding (D-3). Due to these characteristics, we demonstrate that DDF is more suitable to model hand-object interactions. Consequently, DDF-based method achieves more complete and accurate hand-held object reconstruction results (S-4, D-4).

reconstruction, SDF-based methods either directly encode hand pose as a global cue [67] or propagate information between hand and object in 2D feature space [10], which fails to model hand-object interactions in 3D space. **Second**, SDF is compact and can not naturally encapsulate the inherent characteristics of an object such as symmetry. However, many man-made objects in everyday scenes exhibit some degree of (partial-)symmetry, and the inability of SDF to capture this information results in a failure to recover high-quality shapes, especially when the object is heavily occluded by the hand.

To overcome aforementioned challenges, we present DDF-HO, a novel **D**irected **D**istance **F**ield (DDF) based **H**and-held **O**bject reconstruction framework, which takes a single RGB-image as input and outputs 3D model of the target object. In contrast to SDF, DDF maps a ray, comprising an origin and a direction, in 3D space to corresponding DDF values, including a binary visibility signal determining whether the intersection exists and a scalar distance value measuring the distance from origin to target along the sampled direction.

As shown in Fig. 1, we demonstrate the superiority of DDF over SDF in modeling hand-object interactions. For each sampled ray, we collect its features to capture hand-object relationship by combining 2D-3D geometric features via our 2D ray-based feature aggregation and 3D intersection-aware hand pose embedding. We first project the ray onto the image, yielding a 2D ray or a dot (degeneration case), and then aggregate features of all the pixels along the 2D ray as the 2D features, which encapsulate 2D local hand-object cues. Then we collect 3D geometric features, including direct hand pose embedding as global information [67] and ray-hand intersection embedding as local geometric prior. In this manner, hand pose and shape serve as strong priors to enhance the object reconstruction, especially when there is heavy occlusion. Additionally, we also introduce a geometric loss term to exploit the symmetry of everyday objects. In particular, we randomly sample two bijection sets of 3D rays, where corresponding rays have identical origin on the reflective plane but with opposite directions. Thus the DDF predictions of corresponding rays in the two sets should be the same, enabling a direct supervision loss for shape.

In summary, our main contributions are as follows. **First**, we present DDF-HO, a novel hand-held object reconstruction pipeline that utilizes DDF as the shape representation, demonstrating superiority in modeling hand-object intersections over SDF-based competitors. **Second**, we extract local to global features capturing hand-object relationship by introducing a novel 2D ray-based feature aggregation scheme and a 3D intersection-aware hand pose embedding. **Third**, extensive experiments on synthetic and real-world datasets demonstrate that our method consistently outperforms competitors by a large margin, enabling real-world applications requiring high-quality hand-held object reconstruction.

## 2 Related Works

**Hand Pose Estimation.** Hand pose estimation methods from RGB(-D) input can be broadly categorized into two streams: model-free and model-based methods. Model-free methods typically involve lifting detected 2D keypoints to 3D joint positions and hand skeletons [32, 44, 45, 46, 53, 52, 80]. Alternatively, they directly predict 3D hand meshes [11, 21, 48]. On the other hand, model-based methods [3, 55, 58, 77, 79] utilize regression or optimization techniques to estimate statistical models with low-dimensional parameters, such as MANO [54]. Our approach aligns with the model-based stream of methods, as they tend to be more robust to occlusion [67].

**Single-view Object Reconstruction.** The problem of single-view object reconstruction using neural networks has long been recognized as an ill-posed problem. Initially, researchers focus on designing category-specific networks for 3D prediction, either with direct 3D supervision [7, 35, 16] or without it [25, 34, 38, 68, 17]. Some approaches aim to learn a shared model across multiple categories using 3D voxel representations [12, 22, 64, 65, 57], meshes [24, 27, 63, 51], or point clouds [19, 40]. Recently, neural implicit representations have emerged as a powerful technique in the field [42, 31, 2, 1, 47, 33, 70]. These approaches have demonstrated impressive performance.

**Hand-held Object Reconstruction.** Accurately reconstructing hand-held objects presents a significant challenge, yet it plays a crucial role in understanding human-object interaction. Prior works [20, 28, 59, 61] aim to simplify this task by assuming access to known object templates and jointly regressing hand poses and 6DoF object poses [15, 18, 75, 76, 60, 71]. Joint reasoning approaches encompass various techniques, including implicit feature fusion [9, 23, 41, 56], leveraging geometric constraints [4, 6, 13, 26, 74], and encouraging physical realism [62, 49]. Recent researches focus on directly reconstructing hand-held object meshes without relying on any prior assumptions. These methods aim to recover 3D shapes from single monocular RGB inputs. For instance, [29] designs a joint network that predicts object mesh vertices and MANO parameters of the hand, while [36] predicts them in the latent space. Additionally, [10] and [67] utilize Signed Distance Field (SDF) as the representation of hand and object shapes. In contrast, our method introduces a novel representation called Directed Distance Field (DDF) and demonstrates its superiority in reconstructing hand-held objects, surpassing the performance of previous SDF-based methods.

## 3 Method

### 3.1 Preliminaries

**SDF**. Consider a 3D object shape $\mathcal{O} \subset \mathcal{B}$, where $\mathcal{B} \subset \mathbb{R}^3$ denotes the bounding volume that will act as the domain of the field, SDF maps a randomly sampled point $\mathcal{P} \in \mathcal{B}$ to a a scalar value $d$ representing the shortest distance from $\mathcal{P}$ to the surface of the 3D object shape $\mathcal{O}$. This scalar value can be positive, negative, or zero, depending on whether the point lies outside, inside, or on the surface of the object, respectively.

**From SDF to DDF**. SDF is widely used in the object reconstruction, however, due to its inherent undirected and compact nature, it is hard to effectively represent the complex hand-object interactions, as explained in Fig. 1. Hence, in this paper, we propose to utilize DDF, recently proposed and applied by [2, 31, 69], as an extension of SDF for high-quality hand-held object reconstruction.

**DDF**. Given a 3D ray $L_{\mathcal{P},\theta}(t) = \mathcal{P} + t\theta$, consisting of an origin $\mathcal{P} \in \mathcal{B}$ and a view direction $\theta \in \mathbb{S}^2$, where $\mathbb{S}^2$ denotes the set of 3D direction vectors having 2 degree-of-freedom. If this ray intersects with the target object $\mathcal{O} \subset \mathcal{B}$ at some $t \geq 0$, it is considered as *visible*, and DDF maps it to a non-negative scalar field $\mathcal{D} : \mathcal{B} \times \mathbb{S}^2 \rightarrow \mathbb{R}_+$, measuring the distance from the origin $\mathcal{P}$ towards the first intersection with the object along the direction $\theta$. To conveniently model the *visibility* of a ray, a

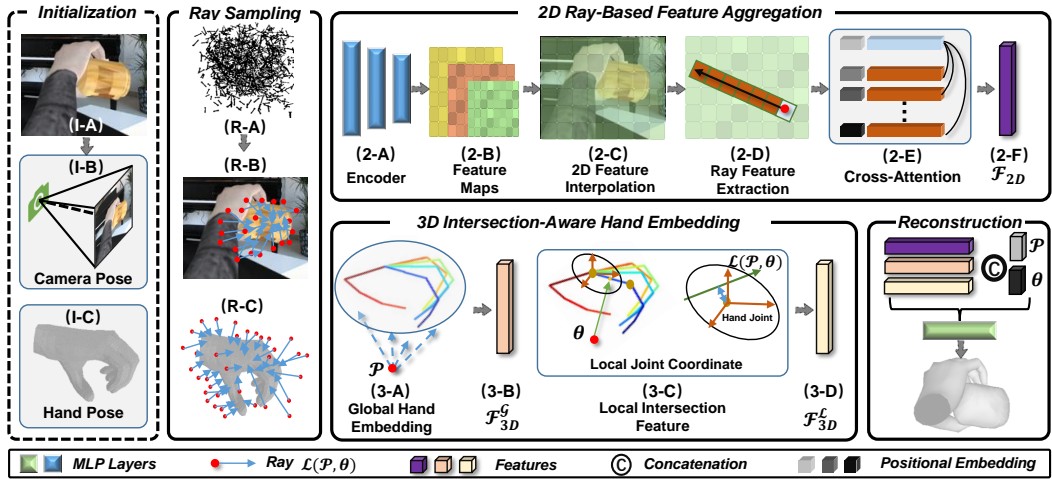

Figure 2: **Overview of DDF-HO.** Given an RGB-image (I-A), we first employ an off-the-shelf pose detector to predict camera pose $\theta_C$ and hand pose $\theta_H$ (45D parameters defined in MANO model [54]), as shown in (I-B) and (I-C) respectively. For the input of DDF, we sample multiple rays (R-A) in 3D space, and project them onto the 2D image (R-B). The corresponding intersections with the hand skeleton are also calculated (R-C). Then for each ray $\mathcal{L}_{\mathcal{P},\theta}$, we collect 2D ray-based feature $\mathcal{F}_{2D}$ from (2-A) to (2-F), and 3D intersection-aware hand embedding $\mathcal{F}_{3D}^{\mathcal{G}}$ and $\mathcal{F}_{3D}^{\mathcal{L}}$ from (3-A) to (3-D). Finally, we concatenate all features and ray representation as $\mathcal{F} = \{\mathcal{P}, \theta, \mathcal{F}_{2D}, \mathcal{F}_{3D}^{\mathcal{G}}, \mathcal{F}_{3D}^{\mathcal{L}}\}$ to predict corresponding DDF values.

binary visibility field is introduced as $\xi(\mathcal{P}, \theta) = \mathbb{1}[L_{\mathcal{P},\theta}$ is visible$]$, *i.e.* for a visible ray, $\xi(\mathcal{P}, \theta) = 1$. Moreover, [2, 31] provide several convenient ways to convert DDF to other 3D representations including point cloud, mesh and vanilla SDF.

## 3.2 DDF-HO: Overview

**Objective**. Given a single RGB image $\mathcal{I}$ containing a human hand grasping an arbitrary object, DDF-HO aims at reconstructing the 3D shape $\mathcal{O}$ of the target object, circumventing the need of object template, category or depth priors.

**Initialization**. As shown in Fig. 2 (I-A)-(I-C), we first adopt an off-the-shelf framework [29, 55] to estimate the hand articulation $\theta_H$ and the corresponding camera pose $\theta_C$ for the input image $\mathcal{I}$, where $\theta_H$ is defined in the parametric MANO model with 45D articulation parameters [54] and $\theta_C$ denotes the 6D pose, rotation $\mathcal{R} \in \mathbb{SO}(3)$ and translation $t \in \mathbb{R}^3$, of the perspective camera with respect to the world frame.

**Image Feature Encoding**. Hierarchical feature maps of image $\mathcal{I}$ are extracted via ResNet [30] to encode 2D cues, as shown in Fig. 2 (2-A) and (2-B).

**Ray Sampling**. We sample 3D rays $\{\mathcal{L}_{\mathcal{P},\theta}\}$, with origins $\mathcal{P} \in \mathcal{B}$ and directions $\theta \in \mathbb{S}^2$, and transform the rays into the normalized wrist frame with the predicted hand pose $\theta_H$, as in IHOI [67]. The specific ray sampling algorithm adopted in the training stage is introduced in detail in the Supplementary Material.

**Ray Feature Aggregation**. To predict corresponding DDF values of $\{\mathcal{L}_{\mathcal{P},\theta}\}$, we collect and concatenate three sources of information: basic ray representations $\{\mathcal{P}, \theta\}$, 2D projected ray features $\mathcal{F}_{2D}$, 3D intersection-aware hand features $\mathcal{F}_{3D}$. For $\mathcal{F}_{2D}$, we project each 3D ray onto the feature maps extracted from $\mathcal{I}$, yielding a 2D ray $\{l_{p,\theta^*}\}$ or a dot (degeneration case). Note that in the degeneration case, the sampled 3D ray passes through the camera center, we only need to collect $\mathcal{F}_{2D}$ inside the patch centered at $p$, as in the SDF-based methods [67]. For other non-trivial cases, we aggregate features along $\{l_{p,\theta^*}\}$ using the 2D Ray-Based Feature Aggregation technique, introduced in detail in Sec. 3.3. For $\mathcal{F}_{3D}$, besides global hand pose embedding as in [67], we also encapsulate the intersection of the ray with the hand joints as local geometric cues to depict the relationship of hand-object interaction, which is further introduced in detail in Sec. 3.4.

**DDF Reconstruction**. Concatenating $\{\mathcal{P}, \theta, \mathcal{F}_{2D}, \mathcal{F}_{3D}\}$ as input, we employ an 8-layer MLP network [31] to predict corresponding DDF values.

### 3.3 Ray-Based Feature Aggregation

Previous SDF-based methods [67] typically aggregates feature for each sampled point within a local patch centered at its projection, posing a significant challenge for hand-held object reconstruction as the aggregated feature may not contain necessary information for predicting the intersection, as illustrated in Fig. 1 (S-2). When the sampled point is far from the object surface, its local feature may even be completely extracted from the background, making it infeasible to predict corresponding SDF values. As a consequence, SDF-based methods either leverage hand pose as a global cue [67] or only propagate hand-object features in 2D space [10], failing to capture hand-object interactions in 3D space. In DDF-HO, besides the ray representation

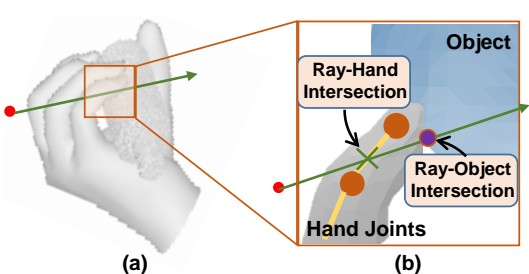

Figure 3: **3D intersection-aware local geometric feature** $\mathcal{F}_{3D}^{\mathcal{L}}$. We collect it by resolving the nearest neighboring hand joints of ray-hand intersection.

$\{\mathcal{P}, \theta\}$, we combine two additional sources of features $\mathcal{F}_{2D}$ and $\mathcal{F}_{3D}$ for each sampled 3D ray to effectively aggregate all necessary information for predicting the DDF value.

We first collect $\mathcal{F}_{2D}$ from the input image $\mathcal{I}$, by employing our 2D Ray-Based Feature Aggregation technique. Given a 3D ray $\mathcal{L}_{\mathcal{P}, \theta}$, as shown in Fig. 2 (R-A) and (R-B), the origin is projected via $p = K(\mathcal{R}\mathcal{P} + t)/\mathcal{P}_z$, where $\mathcal{P}_z$ denotes $z$ component of $\mathcal{P}$, and the direction $\theta^*$ is determined as the normal vector from $p$ towards the projection of another point $\mathcal{P}^*$ on the 3D ray, yielding the projected 2D ray $l_{p, \theta^*}$. Then we sample $K_l$ points $\{p_l^i, i = 1, ..., K_l\}$ on the 2D ray, and extract local patch features $\mathcal{F}_{2D}^l = \{\mathcal{F}^i\}$ for all $K_l$ points as well as the feature $\mathcal{F}_{2D}^p$ of origin projection $p$ via bilinear interpolation on the hierarchical feature maps of $\mathcal{I}$. Finally, we leverage the cross-attention mechanism to aggregate 2D ray feature $\mathcal{F}_{2D}$ for $L_{\mathcal{P}, \theta}$ as,

$$\mathcal{F}_{2D} = \mathcal{F}_{2D}^p + MultiH(\mathcal{F}_{2D}^p, \mathcal{F}_{2D}^l, \mathcal{F}_{2D}^l) \tag{1}$$

where $MultiH$ refers to the multi-head attention and $Q = \mathcal{F}_{2D}^p, K = V = \mathcal{F}_{2D}^l$.

Comparing with single point based patch features used in SDF methods, $\mathcal{F}_{2D}$ naturally captures more information, leading to superior reconstruction quality. First, 2D features from the origin of the 3D ray towards its intersection on the object surface are aggregated, enabling reliable DDF prediction for the ray whose origin is far from the object surface. In this manner, we can sample 3D rays in the whole domain, as shown in Fig. 1 (D-1), encapsulating and reconstructing hand and object simultaneously with a single set of samples. Second, features related to the hand along the projected 2D ray are also considered, providing strong priors for object reconstruction.

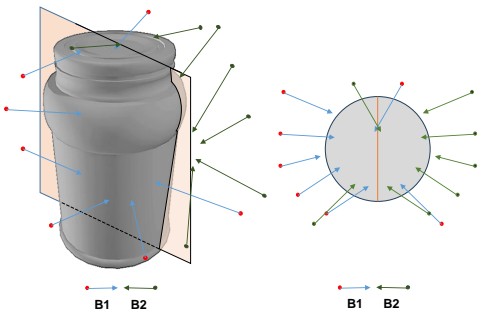

Figure 4: **Construction of the symmetry loss.**

### 3.4 Hand-Object Interaction Modelling

We model hand-object interactions in two aspects. First, in 2D features maps, hand information along the projected 2D ray is encoded into $\mathcal{F}_{2D}$, as introduced in Sec. 3.3. Second, we collect $\mathcal{F}_{3D}$, which encodes both global hand pose embedding $\mathcal{F}_{3D}^{\mathcal{G}}$ as in [67] (details in Supplementary Material) and local geometric feature $\mathcal{F}_{3D}^{\mathcal{L}}$ indicating the intersection of each ray with the hand. As shown in Fig. 2 (3-C) and (3-D), for each 3D ray $\mathcal{L}_{\mathcal{P}, \theta}$, $\mathcal{F}_{3D}^{\mathcal{L}}$ is collected in three steps. First, we calculate the

shortest path from $\mathcal{L}_{\mathcal{P},\theta}$ towards the hand skeleton, constructed by the MANO model and predicted hand articulation parameter $\theta_H$, yielding starting point $\mathcal{P}^{\mathcal{S}}$ on $\mathcal{L}_{\mathcal{P},\theta}$ and endpoint $\mathcal{P}^{\mathcal{D}}$ on the hand skeleton. Then we detect $K_{3D}$ nearest neighboring hand joints of $\mathcal{P}^{\mathcal{D}}$ on the hand skeleton, using geodesic distance. Finally, $\mathcal{P}^{\mathcal{S}}$ is transformed to the local coordinates of detected hand joints (Fig. 2 3-C), indicated by the MANO model, and thereby obtaining $\mathcal{F}_{3D}^{\mathcal{L}}$ by concatenating all these local coordinates of $\mathcal{P}^{\mathcal{S}}$. In summary, $\mathcal{F}_{3D}$ is represented as $\mathcal{F}_{3D} = \{\mathcal{F}_{3D}^{\mathcal{G}}, \mathcal{F}_{3D}^{\mathcal{L}}\}$.

Our hand-object interaction modeling technique has two primary advantages over IHOI [67], in which sampled points are only encoded with all articulation points of the hand skeleton, as shown in Fig. 1 (S-3). First, our technique extracts and utilizes 2D features $\mathcal{F}_{2D}$ that reflect the interaction between the hand and object, providing more useful cues to reconstruct hand-held objects. Second, we incorporate 3D intersection-based hand embeddings $\mathcal{F}_{3D}$ that offer more effective global to local hand cues as geometric priors to guide the learning of object shape, especially when a 3D ray passing through the contact region between the hand and object. In such case, the intersection with the hand skeleton is closer to the intersection with the object than the origin of the ray, thereby encoding the local hand information around the intersection provides useful object shape priors, as shown in Fig. 3. In other cases, our method works similarly to IHOI, where the embeddings serve as a global locator to incorporate hand pose.

### 3.5 Conditional DDF for Hand-held Object Reconstruction

Given concatenated feature $\mathcal{F} = \{\mathcal{P}, \theta, \mathcal{F}_{2D}, \mathcal{F}_{3D}\}$ for each 3D ray $\mathcal{L}_{\mathcal{P},\theta}$, we leverage an 8-layer MLP to map $\mathcal{F}$ to the corresponding DDF value: distance $D$ and binary visible signal $\xi$. The input $\{\mathcal{P}, \theta\}$ is positional encoded by $\gamma$ function as [42]. $\xi$ is output after the 3rd layer to leave the network capacity for the harder distance estimation task. We also introduce a skip connection of the input 3D ray $\{\mathcal{P}, \theta\}$ to the 4th layer to preserve low-level local geometry.

The loss functions of our conditional directed distance field network consist of depth term $\mathcal{L}_D = \xi|\hat{D} - D|$, visibility term $\mathcal{L}_\xi = BCE(\hat{\xi}, \xi)$ and symmetry term $\mathcal{L}_s = |\hat{D}_1 - \hat{D}_2|$, with $\hat{D}, \hat{\xi}$ being the predictions of $D, \xi$ respectively. For symmetric objects, we randomly sample two bijection sets of 3D rays, where corresponding rays have identical origin on the reflective plane but with opposite directions. Specifically, before our experiments, all symmetric objects are preprocessed to be symmetric with respect to the XY plane $\{X = 0, Y = 0\}$. To determine whether the object is symmetric, we first flipping the sampled points $P$ on the object surface w.r.t the XY plane, yielding $P'$. Then we compare the Chamfer Distance between the object surface and $P'$. If the distance lies below a threshold (1e-3), the object is considered symmetric. As for building two bijection sets $B_1 : \{P_1, \theta_1\}$ and $B_2 : \{P_2, \theta_2\}$, we first randomly sample origins $P_1 : \{(x_1, y_1, z_1))\}$ and directions $\theta_1 : \{(\alpha_1, \beta_1, \gamma_1))\}$ to construct $B_1$. Then, we flip $B_1$ with respect to the reflective plane to generate $B_2$ by $P_2 : \{(x_1, y_1, -z_1))\}, \theta_2 : \{(\alpha_1, \beta_1, -\gamma_1))\}$. Since the object is symmetric, the DDF values $\hat{D}_1, \hat{D}_2$ of corresponding rays in $B_1$ and $B_2$ should be the same, which establishes our symmetry loss term. The process of constructing the symmetry loss is shown in Fig. 4.

The final loss is defined as $\mathcal{L} = \mathcal{L}_\xi + \lambda_1 \mathcal{L}_D + \lambda_2 \mathcal{L}_s$, where $\lambda_1, \lambda_2$ are weighting factors.

## 4 Experiments

### 4.1 Experimental Setup

**Datasets.** A synthetic dataset ObMan [29] and two real-world datasets HO3D(v2) [28], MOW [6] are utilized to evaluate DDF-HO in various scenarios. ObMan consists of 2772 objects of 8 categories from ShapeNet [8], with 21K grasps generated by GraspIt [43]. The grasped objects are rendered over random backgrounds through Blender [1]. We follow [29, 67] to split the training and testing sets. HO3D(v2) [28] contains 77,558 images from 68 sequences with 10 different persons manipulating 10 different YCB objects [5]. The pose annotations are yielded by multi-camera optimization pipelines. We follow [28] to split training and testing sets. MOW [6] comprises a total of 442 images and 121 object templates, collected from in-the-wild hand-object interaction datasets [14, 56]. The approximated ground truths are generated via a single-frame optimization method [6]. The training and testing splits remain the same as the released code of [67].

---

[1]https://www.blender.org/

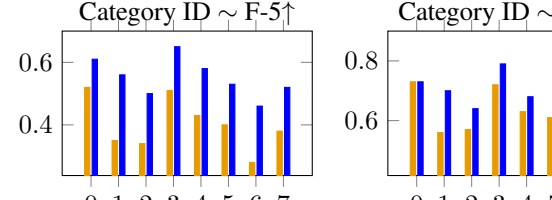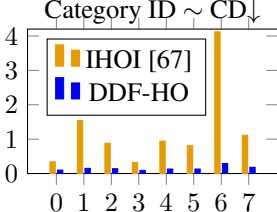

Figure 5: Reconstruction quality comparisons on Obman [29] by category. From 0 to 7, we demonstrate the results of IHOI [67] *vs* DDF-HO on the categories: Bottle, Bowl, Camera, Can, Cellphone, Jar, Knife, Remote Control sequentially. DDF-HO consistently outperforms IHOI.

| Method | F-5 ↑ | F-10 ↑ | CD ↓ | F-5 ↑ | F-10 ↑ | CD ↓ |
|---|---|---|---|---|---|---|
| HO [29] | 0.08 | 0.19 | 4.60 | 0.05 | 0.14 | 6.03 |
| GF [36] | 0.09 | 0.21 | 5.23 | 0.07 | 0.16 | 6.25 |
| IHOI [67] | 0.21 | 0.38 | 1.99 | 0.17 | 0.31 | 4.17 |
| Ours | **0.28** | **0.42** | **0.55** | **0.24** | **0.36** | **0.73** |

Table 1: Results on HO3D(v2) [28] dataset with finetuning (left) and zero-shot generalization from ObMan [29] dataset (right). Overall best results are **in bold**.

**Evaluation Metrics.** We first utilize [2] to convert the predicted DDF into point cloud representation and then compare against the sampled point cloud from the corresponding ground truth mesh. Following [67], we report Chamfer Distance (CD, mm), F-score at 5mm threshold (F-5) and 10mm threshold (F-10) of the converted object point cloud, which reflects the quality of hand-held object reconstruction.

**Baselines.** Three baseline methods are chosen to ensure fair comparisons with our DDF-HO method, taking into account that they aim to predict the shape of hand-held objects directly from a single RGB input and do not rely on known object templates, category information, or depth priors. The baselines include Atlas-Net [27] based HO [29], implicit field based GF [36] and SDF based IHOI [67].

**Implementation Details.** We conduct the training, evaluation and visualization of DDF-HO on a single A100 40GB GPU. We use the same off-line systems [29, 55] as [67] to estimate the hand and camera poses. We also adopt a ResNet34 [30] image encoder to extract a 5-resolution visual feature pyramid, the same with [67]. In all the three datasets, we sample 20K 3D rays for each object and generate the ground truth using the ray marching algorithm in Trimesh [2]. The number of sampled points $K_l$ along the projected 2D ray is set to 8 and number of multi-head attention is 2 for 2D Ray-Based Feature Aggregation technique. $K_{3D}$ for $\mathcal{F}_{3D}^{\mathcal{L}}$ introduced in Sec. 3.4 is set as 8. DDF-HO is trained end-to-end using Adam with a learning rate of 1e-4 on ObMan for 100 epochs. Following [67], we use the network weights learned on synthetic ObMan to initialize the training on HO3D(v2) and MOW. Training on HO3D(v2) and MOW also use Adam optimizer with a learning rate 1e-5 for another 100 and 10 epochs, respectively following [67]. The weighting factors of the loss for DDF-HO $\lambda_1, \lambda_2$ are set to 5.0 and 0.5, respectively. Note that we do not include the symmetry loss term $\mathcal{L}_s$ in the training except in the Ablation Study Tab. 4 for a fair comparison, since other baselines do not leverage this additional information. Note that during evaluation, we convert the DDF representation to point cloud [2], while during visualization, we convert DDF to mesh [2, 31]. Details of ground truth generation and network architecture are provided in the Supplementary Material.

### 4.2 Evaluation on Synthetic Scenarios

As shown in Tab. 3, the evaluation on the synthetic large-scale dataset ObMan demonstrates that DDF-HO achieves high-quality hand-held object reconstruction with the assistance of suitable DDF representation. Our proposed method exhibits state-of-the-art performance under all three evaluation metrics of F-5, F-10, and CD. Specifically, we achieve a significant improvement over the current state-of-the-art SDF-based IHOI, with a gain of 13% and 4% on F-5 and F-10, respectively. This improvement can be attributed to DDF-HO's more suitable ray-based feature aggregation technique,

---

[2]https://trimsh.org/

| Method | F-5 ↑ | F-10 ↑ | CD ↓ | F-5 ↑ | F-10 ↑ | CD ↓ |
|---|---|---|---|---|---|---|
| IHOI [67] | 0.10 | 0.19 | 7.83 | 0.09 | 0.17 | 8.43 |
| Ours | **0.16** | **0.22** | **1.59** | **0.14** | **0.19** | **1.89** |

Table 2: Results on MOW [6] dataset, with the setting of finetuning (left) and zero-shot generalization from ObMan [29] dataset (right). Overall best results are **in bold**.

Figure 6: **Visualization results on ObMan [29].** Our method consistently outperforms IHOI [67].

which allows for more exquisite hand-held object reconstruction. Notably, DDF-HO's CD metric is reduced by 85% compared to IHOI and 77% compared to HO, indicating that our predicted object surface contains much fewer outliers.

Fig. 6 presents improved visualizations of DDF-HO, showcasing enhanced and more accurate surface reconstruction of hand-held objects. While IHOI can achieve decent object surface recovery within the camera view, the reconstructed surface appears rough with numerous outliers when observed from novel angles. This suggests that IHOI lacks the ability to perceive 3D hand-held objects due to its limited modeling of hand-object interaction in 3D space. In contrast, DDF-HO utilizes a more suitable DDF representation, resulting in smooth and precise reconstructions from any viewpoint of the object.

### 4.3 Evaluation on Real-world Scenarios

In addition to synthetic scenarios, we conduct experiments on two real-world datasets, HO3D(v2) and MOW, to evaluate DDF-HO's performance in handling real-world human-object interactions . Table 1 constitutes the evaluation results on HO3D(v2) after finetuning (with related settings described in Section 4.1), as well as the zero-shot generalization results, where we directly conducted inference on HO3D(v2) using the weights from training on ObMan. Furthermore, Table 2 presents results on MOW under the same setting.

| Method | F-5 ↑ | F-10 ↑ | CD ↓ |
|---|---|---|---|
| HO [29] | 0.23 | 0.56 | 0.64 |
| GF [36] | 0.30 | 0.51 | 1.39 |
| IHOI [67] | 0.42 | 0.63 | 1.02 |
| Ours | **0.55** | **0.67** | **0.14** |

Table 3: Results on ObMan [29] dataset. Overall best results are **in bold**.

Our method, after finetuning on the real-world data, achieves state-of-the-art performance on both datasets. Specifically, on HO3D(v2), we observe a considerable improvement compared to IHOI and other methods, with an increase in F-5 by 7%, F-10 by 4%, and a significant decrease in CD by 73%. On MOW, our approach also outperforms the previous state-of-the-art methods, achieving a remarkable performance gain in terms of increased F-5 (6%), F-10 (3%), and reduced CD (80%) compared with IHOI [67]. Fig. 7 shows the visualization comparison results on MOW. DDF-HO performs well under the real-world scenario, yielding more accurate reconstruction results.

Furthermore, the zero-shot experiments demonstrate that DDF-HO has a stronger ability for synthetic-to-real generalization. Specifically, on HO3D(v2), DDF-HO yields superior performance in terms of F-5 by 7%, F-10 by 5%, and a decreased CD by 82%. Moreover, the results on MOW also indicate

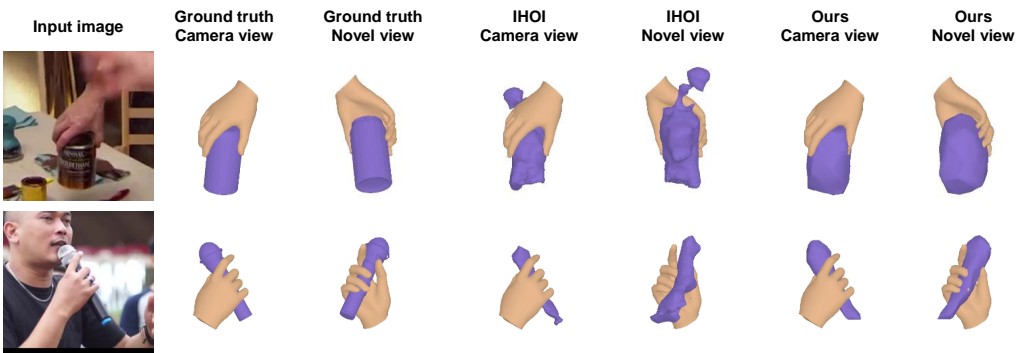

Figure 7: **Visualization results on MOW [6].** Our method consistently surpasses IHOI [67].

that our method, trained only on synthetic datasets, can still achieve decent performance in real-world scenarios, thanks to the generic representation ability of DDF for hand-object interaction modeling.

## 4.4 Efficiency

To demonstrate the efficiency of DDF-HO, we compare the network size and the running speed with IHOI, which is a typical SDF-based hand-held object reconstruction method. All experiments are conducted on a single NVIDIA A100 GPU.

DDF-HO runs at 44FPS, which is slower than IHOI with 172 FPS, but still achieves real-time performance. The slower inference comes from the attention calculation for 2D ray-based feature aggregation and the more elaborated 3D feature generation. For model size, the two methods share a similar scale (24.5M of DDF-HO and 24.0M of IHOI). Generally, the increased parameters mainly come from the cross attention mechanism in 2D ray-based feature aggregation. Other modules are only adopted to collect features to model hand-object interactions and do not significantly increase the network size.

## 4.5 Ablation Studies

We conduct ablation studies on the ObMan and HO3D(v2) datasets to evaluate the impact of three key assets of DDF representation: Ray-Based Feature Aggregation (RFA), Intersection-aware Hand Feature (IHF), and Symmetry Loss (SYM). The results of the ablation studies are presented in Tab. 4.

On ObMan, we first replace the SDF representation with DDF without any modifications to feature aggregation, resulting in a slight improvement over the SDF-based IHOI with a $3\%$ increase in F-5 metric. This indicates that although DDF is more suitable for representing hand-object interaction (with almost an $80\%$ decrease in CD metric), more sophisticated feature aggregation designs are required. Next, we add RFA considering the characteristics of sampled rays, leading to a $5\%$ increase in F-5 and a $4\%$ increase in F-10. Subsequently, adding IHF, which models hand-object interaction locally by considering the intersection information of the hand, resulted in a $5\%$ increase in F-5 and a $3\%$ increase in F-10. This indicates that considering the intersection information of the hand can improve the accuracy of hand-held object reconstruction. Finally, adding the SYM loss, which captures the symmetry nature of everyday objects and handles self-occluded scenarios caused by hands, results in another $2\%$ increase in F-5 and a $1\%$ increase in F-10. On the HO3D(v2) dataset, RFA, IHF, and SYM modules play similar roles as on ObMan.

Additionally, we evaluate the influence of input hand pose on DDF-HO by adding Gaussian noise to the estimated hand poses (Pred) or ground truth hand poses (GT) in the input. The results of the ablation studies (Tab. 5) demonstrate the robustness of DDF-HO in handling noisy input hand poses.

## 5 Conclusion

In this paper, we present DDF-HO, a novel pipeline that utilize DDF as the shape representation to reconstruct hand-held objects, and demonstrate its superiority in modeling hand-object interactions

| RFA | IHF | SYM | F-5 ↑ | F-10 ↑ | CD ↓ | F-5 ↑ | F-10 ↑ | CD ↓ |
|---|---|---|---|---|---|---|---|---|
| SDF baseline [67] | | | 0.42 | 0.63 | 1.02 | 0.21 | 0.38 | 1.99 |
| ✗ | ✗ | ✗ | 0.45 | 0.60 | 0.22 | 0.24 | 0.38 | 0.62 |
| ✓ | ✗ | ✗ | 0.50 | 0.64 | 0.17 | 0.26 | 0.39 | 0.58 |
| ✓ | ✓ | ✗ | 0.55 | 0.67 | 0.14 | 0.28 | 0.42 | 0.55 |
| ✓ | ✓ | ✓ | **0.57** | **0.68** | **0.13** | **0.29** | **0.43** | **0.52** |

Table 4: Ablation studies for key assets of DDF-HO on ObMan [29] (left) and HO3D(v2) [28] datasets (right).

| Noise | F-5 ↑ | F-10 ↑ | CD ↓ | F-5 ↑ | F-10 ↑ | CD ↓ |
|---|---|---|---|---|---|---|
| Pred | 0.55 | 0.67 | 0.14 | 0.28 | 0.42 | 0.55 |
| Pred+$\sigma = 0.1$ | 0.54 | 0.66 | 0.15 | 0.24 | 0.35 | 0.73 |
| Pred+$\sigma = 0.5$ | 0.47 | 0.60 | 0.18 | 0.20 | 0.30 | 0.83 |
| Pred+$\sigma = 1.0$ | 0.42 | 0.55 | 0.25 | 0.17 | 0.27 | 0.98 |
| Pred+$\sigma = 1.5$ | 0.38 | 0.52 | 0.31 | 0.14 | 0.25 | 1.24 |
| GT | 0.59 | 0.70 | 0.10 | 0.30 | 0.45 | 0.50 |
| GT+$\sigma = 0.1$ | 0.58 | 0.69 | 0.11 | 0.27 | 0.43 | 0.58 |
| GT+$\sigma = 0.5$ | 0.51 | 0.63 | 0.16 | 0.23 | 0.34 | 0.76 |

Table 5: Ablation studies for input hand pose on ObMan [29] (left) and HO3D(v2) [28] datasets (right). The Pred row remains the same setting with the Tab. 1 and 3.

over competitors. Specifically, for each sampled ray in 3D space, we collect its features capturing local-to-global hand-object relationships by introducing a novel 2D ray-based feature aggregation and 3D intersection-aware hand pose embedding. Extensive experiments on synthetic dataset Obman and real-world datasets HO3D(v2) and MOW verify the effectiveness of DDF-HO on reconstructing high-quality hand-held objects.

**Limitations.** DDF-HO naturally inherits the shortcomings of DDF. First, the higher dimensional input of DDF makes it harder to train than SDF, resulting in more complex data, algorithm and network structure requirements. This may hinder the performance of DDF-based methods when scaling up to large-scale scenes, like traffic scenes. Second, to enable more photorealistic object reconstruction, there are other characteristics like translucency, material and appearance need to be properly represented. This requires further research to fill the gap.

**Acknowledgements.** This work was supported by the National Key R&D Program of China under Grant 2018AAA0102801, National Natural Science Foundation of China under Grant 61827804. We also appreciate Yufei Ye, Tristan Aumentado-Armstrong, Zerui Chen and Haowen Sun for their insightful discussions.

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
