# Supplementary Material for DDF-HO: Hand-Held Object Reconstruction via Conditional Directed Distance Field

## 1 Network Architecture

Given an RGB input, DDF-HO first uses a ResNet34 [6] to generate hierarchical feature maps at the resolution of 56x56, 28x28, 14x14 and 7x7 respectively (Fig. 1 (a)). We use bilinear interpolation to upsample the feature map to the same size, which is 56x56. Then, all the upsampled feature maps go through a bottleneck convolutional layer to have a channel size of 256, resulting in an output size of 56x56x256. After 2D ray sampling process depicted in Sec. 3.3 in the main manuscript, the local point feature $\mathcal{F}_{2D}^p$ ($Q$) and the sampled local ray feature $\mathcal{F}_{2D}^l$ ($K$ and $V$) are fed to a multi-head attention network [9] with number of head 2 during our implementation. The aggregated 2D ray feature $\mathcal{F}_{2D}$ is generated by the addition of $\mathcal{F}_{2D}^p$ and the output of the multi-head attention layer (Fig. 1 (b)). The concatenated feature $\mathcal{F} = \{\mathcal{P}, \theta, \mathcal{F}_{2D}, \mathcal{F}_{3D}\}$ for each 3D ray $\mathcal{L}_{\mathcal{P}, \theta}$ is fed to an 8-layer MLP to yield corresponding DDF value: distance $D$ and binary visible signal $\xi$, as described in Sec. 3.5 in the main manuscript (Fig. 1 (c). The total number of parameters of our network is 25M. Thanks to the simple design of our network architecture, DDF-HO can reconstruct object in real-time ($\sim$ 50 FPS).

## 2 Details of 3D Intersection-aware Hand Features

In Section 3.4 of the main manuscript, we introduce the 3D intersection-aware hand feature $\mathcal{F}_{3D}$, which is represented as $\mathcal{F}_{3D} = \{\mathcal{F}_{3D}^{\mathcal{G}}, \mathcal{F}_{3D}^{\mathcal{L}}\}$. The specific construction of $\mathcal{F}_{3D}^{\mathcal{L}}$ is detailed in Section 3.4. The global hand pose embedding $\mathcal{F}_{3D}^{\mathcal{G}}$ is created using an explicit articulation embedder based on the approach described in [11]. To elaborate, given the hand articulation parameter $\theta_H$, we employ forward kinematics [11, 8] to derive the transformation that maps a sample point $\mathcal{P}$ to each joint coordinate. The resulting 15 joint coordinates corresponding to $\mathcal{P}$ are concatenated to form $\mathcal{F}_{3D}^{\mathcal{G}}$. This process enables the extraction of global information from the hand joints.

## 3 Ray Sampling Algorithm for DDF

During the testing stage, the starting points $\{\mathcal{P}\}$ of the sampled rays for DDF-HO are uniformly distributed within a uniform sphere, and the directions $\{\theta\}$ are also uniformly sampled. However, during the training stage, as we have access to the model information, employing diverse sampling algorithms can enhance the training process, leading to a robust 3D perception capability [1].

Our training process involves the utilization of five distinct types of data samples. To sample $\{\mathcal{L}_{\mathcal{P}, \theta}\}$, instead of directly obtaining $\{\theta\}$, we select the corresponding endpoints $\{\mathcal{P}_E\}$ and calculate each $\theta$ as $\theta = (\mathcal{P}_E - \mathcal{P})/|\mathcal{P}_E - \mathcal{P}|$. The five types of data samples for training are as follows: **I.** Both $\{\mathcal{P}\}$ and $\{\mathcal{P}_E\}$ are uniformly sampled within a uniform sphere. **II.** $\{\mathcal{P}\}$ is uniformly sampled within a uniform sphere, while $\{\mathcal{P}_E\}$ is uniformly sampled on the mesh surface. **III.** $\{\mathcal{P}\}$ is sampled along the tangent space of the mesh, while $\{\mathcal{P}_E\}$ is uniformly sampled on the mesh surface. **IV.** $\{\mathcal{P}\}$ is sampled along the tangent space of the mesh and then perturbed with Gaussian noise, while $\{\mathcal{P}_E\}$ is uniformly sampled on the mesh surface. **V.** $\{\mathcal{P}\}$ is sampled on the surface of a uniform sphere, while

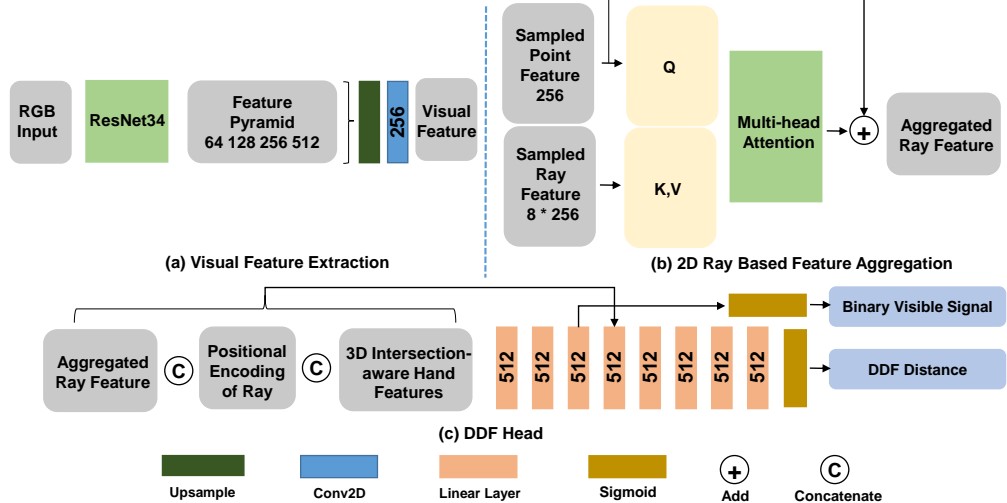

Figure 1: Network Architecture of DDF-HO.

| Method | F-5 ↑ | F-10 ↑ | CD ↓ | F-5 ↑ | F-10 ↑ | CD ↓ |
|---|---|---|---|---|---|---|
| Ours | 0.28 | 0.42 | 0.55 | 0.24 | 0.36 | 0.73 |
| Error Bar | ±0.012 | ±0.010 | ±0.018 | ±0.006 | ±0.005 | ±0.013 |

Table 1: Error bars on HO3D(v2) [4] dataset with finetuning (left) and zero-shot generalization from ObMan [5] dataset (right).

$\{\mathcal{P}_E\}$ is uniformly sampled on the mesh surface. For each object, we sampled a total of 20,000 rays, distributed among the five algorithms as follows: 10,000 for algorithm I, 2,500 for algorithms II, III, IV, and V respectively. The effectiveness of this sampling strategy in training the DDF network has been demonstrated by [1].

## 4 Ground Truth Generation for DDF

Once we have obtained the samples $\{\mathcal{L}_{\mathcal{P},\theta}\}$, we generate the ground truth distance $D$ and binary visible signal $\xi$ using the Trimesh library [1]. The ray marching algorithm in Trimesh is utilized to determine the intersection location of each ray, in the event that it intersects with the object mesh. Simultaneously, Trimesh provides the signal ($\xi$) indicating whether the ray intersects the mesh or not. To calculate the distance $D$, we measure the Euclidean distance between the origin of the ray and its corresponding intersection location along the given direction.

## 5 Error Bars of Experiments

To fully illustrate the statistical significance of our experiments, we provide corresponding error bars of Tab. 1, 2 and 3 in our main manuscript, as is shown in Tab. 1, 2 and 3. We obtain the error bars by running the experiments 5 times with different random seeds.

## 6 Quantitative Results on ObMan by Category

In this section, we provide the corresponding table (Tab. 4) for Fig. 4 in the main manuscript to give more detailed comparison results on ObMan by category. DDF-HO consistently outperforms the

---

[1]https://trimsh.org/

| Method | F-5 ↑ | F-10 ↑ | CD ↓ | F-5 ↑ | F-10 ↑ | CD ↓ |
|---|---|---|---|---|---|---|
| Ours | 0.16 | 0.22 | 1.59 | 0.14 | 0.19 | 1.89 |
| Error Bar | ±0.010 | ±0.006 | ±0.017 | ±0.008 | ±0.010 | ±0.009 |

Table 2: Error bars on MOW [2] dataset, with the setting of finetuning (left) and zero-shot generalization from ObMan [5] dataset (right).

| Method | F-5 ↑ | F-10 ↑ | CD ↓ |
|---|---|---|---|
| Ours | 0.55 | 0.67 | 0.14 |
| Error Bar | ±0.012 | ±0.005 | ±0.003 |

Table 3: Error bars on ObMan [5] dataset.

current state-of-the-art [11] on each category, especially for some tiny objects hard to reconstruct well such as cellphone and knife.

# 7   Additional Quantitative Comparisons on ObMan

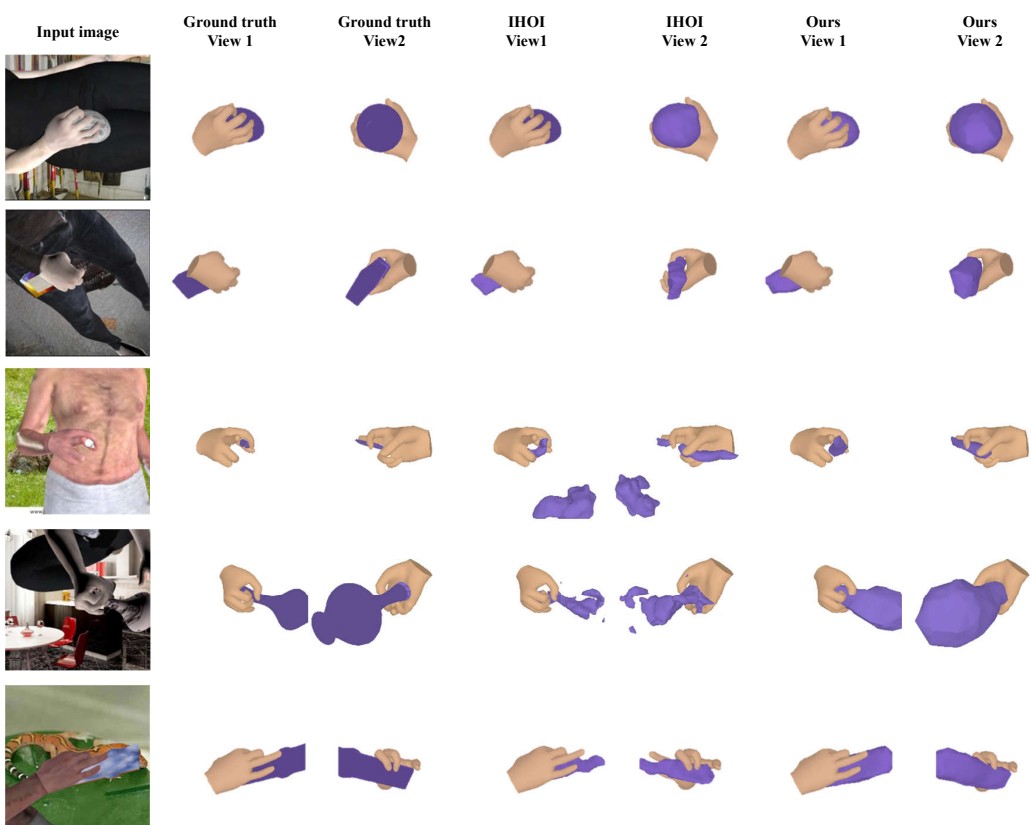

Figure 2: Visualization results on ObMan [5].

Tab. 5 exhibits additional quantitative comparison results on ObMan dataset. We add another recent hand-object reconstruction method AlignSDF [3]. Results show that DDF-HO still consistently outperforms all competitors.

| Metrics | Methods | Bottle | Bowl | Camera | Can | Cellphone | Jar | Knife | Remote | Mean |
|---------|---------|--------|------|--------|-----|-----------|-----|-------|--------|------|
| F-5 ↑ | IHOI [11] | 0.52 | 0.35 | 0.34 | 0.51 | 0.43 | 0.40 | 0.28 | 0.38 | 0.42 |
| | Ours | **0.61** | **0.56** | **0.50** | **0.65** | **0.58** | **0.53** | **0.46** | **0.52** | **0.55** |
| F-10 ↑ | IHOI [11] | **0.73** | 0.56 | 0.57 | 0.72 | 0.63 | 0.61 | 0.46 | 0.59 | 0.63 |
| | Ours | **0.73** | **0.70** | **0.64** | **0.79** | **0.68** | **0.66** | **0.56** | **0.63** | **0.67** |
| CD ↓ | IHOI [11] | 0.34 | 1.54 | 0.88 | 0.32 | 0.94 | 0.81 | 4.12 | 1.11 | 1.02 |
| | Ours | **0.10** | **0.15** | **0.14** | **0.09** | **0.13** | **0.13** | **0.29** | **0.18** | **0.14** |

Table 4: Metrics on ObMan [5] dataset by category. Overall best results are **in bold**.

| Method | F-5 ↑ | F-10 ↑ | CD ↓ |
|--------|-------|--------|------|
| HO [5] | 0.23 | 0.56 | 0.64 |
| GF [7] | 0.30 | 0.51 | 1.39 |
| AlignSDF [3] | 0.40 | 0.64 | 3.38 |
| IHOI [11] | 0.42 | 0.63 | 1.02 |
| Ours | **0.55** | **0.67** | **0.14** |

Table 5: Additional Results on ObMan [5] dataset. Overall best results are **in bold**.

# 8  Additional Qualitative Results

Additional visualization results are shown in Fig. 2, 3, 4. More 3D visualization results are provided in our attached video.

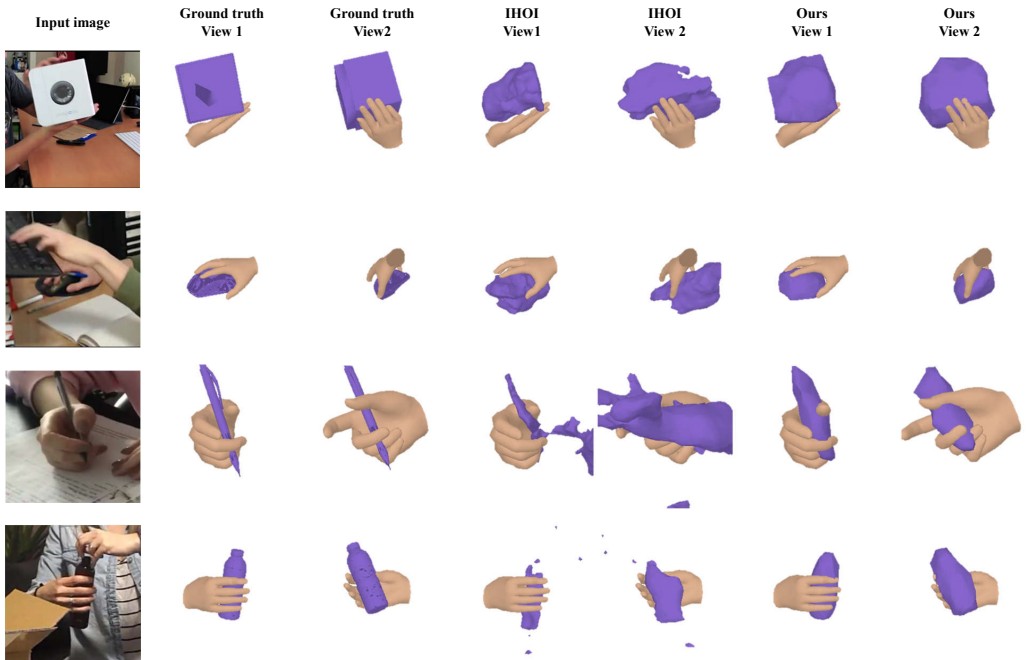

Figure 3: Visualization results on MOW [2].

# 9  Experiments Using Other Representation

We also consider leveraging occupancy as the shape representation to fully demonstrate the superiority of DDF representation for hand-held object reconstruction. Since no off-the-shelf hand-held object reconstruction pipelines leverage occupancy as the shape representation, we design a baseline method ourselves following the widely-used 2D-3D lifting scheme in single-view reconstruction [10]. We first use the same backbone as our method (ResNet34) to extract per-pixel features. The extracted

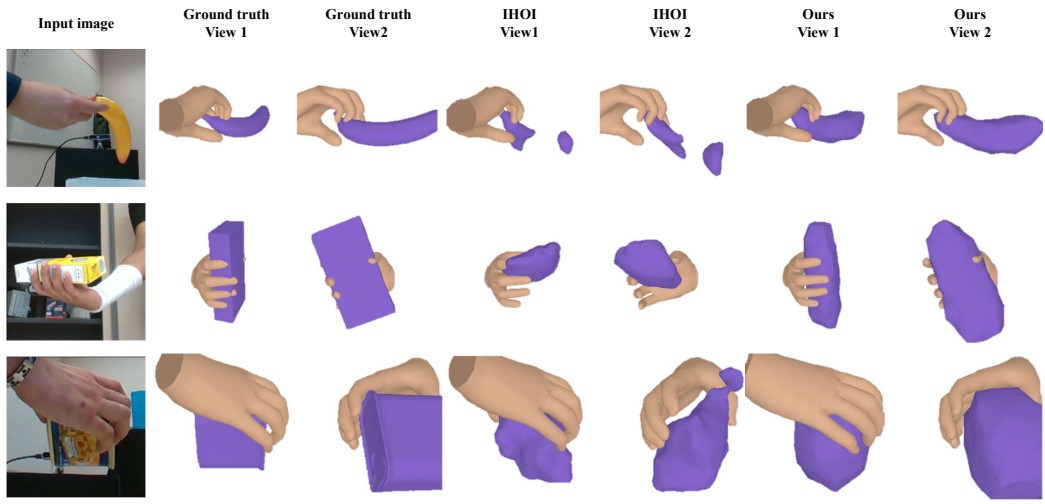

Figure 4: Visualization results on HO3D(v2) [4].

| Method | F-5 ↑ | F-10 ↑ | CD ↓ | F-5 ↑ | F-10 ↑ | CD ↓ |
|---|---|---|---|---|---|---|
| Pix2Vox [10] | 0.24 | 0.45 | 1.81 | 0.06 | 0.17 | 6.12 |
| IHOI [11] | 0.42 | 0.63 | 1.02 | 0.21 | 0.38 | 1.99 |
| Ours | **0.55** | **0.67** | **0.14** | **0.28** | **0.42** | **0.55** |

Table 6: Results considering occupancy representation on ObMan [5] dataset (left) and HO3D [4] dataset (right). Overall best results are **in bold**.

features are then back-projected to the volume (32x32x32, the same as [10]). For each voxel inside the volume, the predicted hand pose (the same as used in DDF-HO, parameterized as the MANO model parameters) is also concatenated to its feature vector. Finally, we predict the occupancy as in Pix2Vox [10]. The results on ObMan (the left block) and HO3D(V2) (the right block) are shown as Tab. 6.

## 10   Failure Cases

We add some visual results of the failure cases to comprehensively showcase our method, as exhibited in Fig. 5. It can be seen that reconstructing very thin objects is still a big challenge for all methods.

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

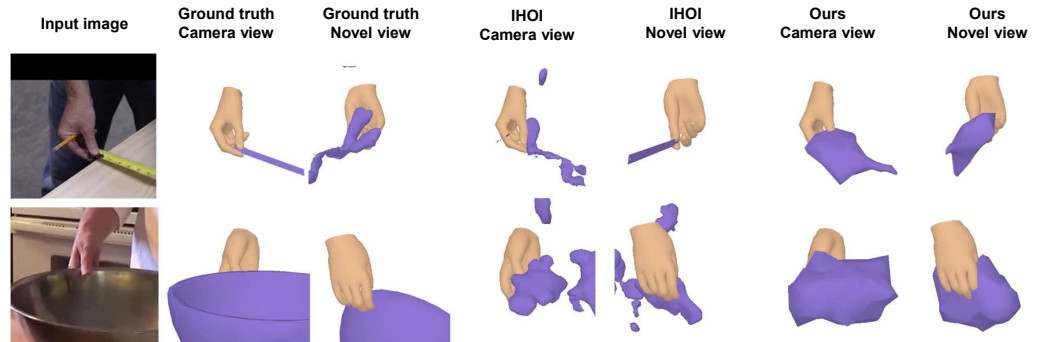

| Input image | Ground truth Camera view | Ground truth Novel view | IHOI Camera view | IHOI Novel view | Ours Camera view | Ours Novel view |

Figure 5: Visualization results of the failure cases
.

[7]  Korrawe Karunratanakul, Jinlong Yang, Yan Zhang, Michael J Black, Krikamol Muandet, and Siyu Tang. Grasping field: Learning implicit representations for human grasps. In *2020 International Conference on 3D Vision (3DV)*, pages 333–344. IEEE, 2020.

[8]  Javier Romero, Dimitrios Tzionas, and Michael J Black. Embodied hands: Modeling and capturing hands and bodies together. *arXiv preprint arXiv:2201.02610*, 2022.

[9]  Ashish Vaswani, Noam Shazeer, Niki Parmar, Jakob Uszkoreit, Llion Jones, Aidan N Gomez, Łukasz Kaiser, and Illia Polosukhin. Attention is all you need. *Advances in neural information processing systems*, 30, 2017.

[10]  Haozhe Xie, Hongxun Yao, Xiaoshuai Sun, Shangchen Zhou, and Shengping Zhang. Pix2vox: Context-aware 3d reconstruction from single and multi-view images. In *Proceedings of the IEEE/CVF international conference on computer vision*, pages 2690–2698, 2019.

[11]  Yufei Ye, Abhinav Gupta, and Shubham Tulsiani. What's in your hands? 3d reconstruction of generic objects in hands. In *Proceedings of the IEEE/CVF Conference on Computer Vision and Pattern Recognition*, pages 3895–3905, 2022.