# OpenReview forum: "DDF-HO: Hand-Held Object Reconstruction via Conditional Directed Distance Field"
_NeurIPS.cc/2023/Conference — NeurIPS 2023 poster_

### Official Review · Reviewer_hGDA · 2023-07-03

**Soundness:** 3 good
**Presentation:** 3 good
**Contribution:** 3 good
**Rating:** 6
**Confidence:** 3

**Summary:**

The paper presents DDF-HO, a method for handheld object reconstruction based on Directed Distance Fields. Given a single RGB image containing a hand grasping an arbitrary object, DDF-HO reconstructs the object without requiring a template or depth priors. Previous methods addressing this problem have relied on the Signed Distance Field (SDF) for the same purpose. The paper includes experiments comparing DDF-HO with such methods on multiple datasets demonstrating the advantages of the proposed method in terms of accuracy.


**Final Rating**

After reading the numerous reviews and the responses of the authors, I believe that this is a paper that can/should be accepted. As I wrote in my previous post, thorough rewriting is needed in some parts to improve clarity.

**Strengths:**

S1. DDF-HO obtains 3D reconstructions of much higher quality than recent methods [25, 32, 62]. The paper provides sufficient evidence that this is due to the use of DDF instead of SDF as the representation, the use of a ray-based feature aggregation scheme, and a 3D intersection-aware hand pose embedding. These contributions lead to significant increases in reconstruction accuracy.

S2. The method is described clearly and with sufficient detail to enable reasonable reproduction. The code is also included, but I did not try to work with it, or identify the key pieces. Handling of symmetry is the exception to this comment.

S3. Three large-scale, widely used datasets, one synthetic and two real, are used to generate the experimental results. The protocol follows that of IHOI [62], which is the most closely related prior work. Two additional recent baseline methods have been chosen for the experiments.

DDF-HO outperforms the baselines on all three datasets. It also shows good zero-shot generalization properties, suggesting that it does not overfit. The experimental results section includes some analysis of potential reasons for the differences in performance across algorithms. The ablation studies are also informative, especially the comparisons to IHOI.

**Weaknesses:**

W1. My primary concern about the paper is the number and complexity of steps. DDF-HO requires sampling rays, computing and aggregating 2D and 3D features, measuring geodesic distances etc. The authors acknowledge that their method has higher complexity than the baselines in the limitations section, but I would like to see more analysis and data on the tradeoffs between speed and accuracy. How long do training and inference take for IHOI and DDF-HO on similar hardware and data? Is interactive deployment feasible?

W2. Handling of symmetric objects is not presented clearly overall. How is the reflective plane of symmetric objects discovered?


Minor Comments (not affecting my recommendation)

I find the use of “undirected” as a property of SDF confusing. The sign and the distance value of SDF direct us to the nearest surface. I do not have a good suggestion of an alternative.

Small language errors can be found throughout the paper. Examples include missing articles and minor inconsistencies.

98: “Arbitrarily” is more appropriate than “randomly.”

Figure 4 and Tables 1 and 2 can be placed closer to the text that refers to them.

Refereed, rather than arXiv, versions of papers should be cited whenever possible.

The first paragraph of Section 3 of the supplement is very important for understanding the algorithm, in my opinion. I suggest finding some space for it in the main paper. One of the weaknesses I had noted was lack of clarity on the ray sampling process.

**Questions:**

Answers to the first two weaknesses listed above could go a long way toward improving my rating. (I went with a conservative rating at this stage.)

**Limitations:**

More information on complexity and run times would have been useful.

---

> ### Author Rebuttal · Authors · 2023-08-08
>
> Thank you for the constructive review. Below we try our best to address your concerns and questions. We have already revised the paper according to the reviews.
>
> ## Q1. Efficiency and Model Complexity of DDF-HO
> IHOI is a typical SDF-based hand-held object reconstruction method. We list detailed comparisons with IHOI in the following table.
>
> ||DDF|IHOI
> |------|------|------|
> Training time| 2d 8h |IHOI: 2d|
> Parameters|24.5M| 24.0M
> Running Speed|23ms/img (44FPS) |5.8ms/img (172FPS)|
>
> From the above table, it can be seen that our method runs slower than IHOI but still achieves real-time performance. All experiments in the above table are conducted on a single NVIDIA A100 GPU. The slower inference comes from the attention calculation for 2D ray-based feature aggregation and the more elabored 3D feature generation. Ray sampling in the testing time only takes less than 1ms since only randomly sampling on unit sphere for ray starting points and directions is conducted. For model size, the two methods share a similar scale. Generally, the increased parameters mainly come from the cross attention mechanism in 2D ray-based feature aggregation (Sec. 3.3). Other modules are only adopted to collect features to model hand-object interactions and do not significantly increase the network size. Moreover, since the network needs to predict both the distance and visibility signal, the convergence speed of DDF-HO during training is also slower than IHOI.
>
> ## Q2. Symmetry Loss
> Since other baseline methods don’t leverage this prior knowledge, we do not use the symmetry loss except in the ablation studies in Tab. 4, as explained in L.243 in the manuscript. Therefore, our results on different datasets and visualization results are all obtained without using the symmetry loss. From Tab. 4, we demonstrate that this symmetry loss can improve the results slightly.
>
> Before our experiments, all symmetric objects in the datasets are preprocessed to be symmetric with respect to the XY plane $\{x=0, y=0\}$ . To determine whether the object is symmetric, we first flipping the sampled points P on the object surface w.r.t the XY plane, yielding P'. Then we compare the Chamfer Distance between the object surface and P'. If the distance lies below a threshold (1e-3), then the object is considered symmetric. As for building the two bijection sets B1 $\{P_1, \theta_1\}$ and B2 $\{P_2, \theta_2\}$, we first randomly sample origins $\{P_1: (x_1, y_1, z_1)\}$ and directions $\{\theta_1: (\alpha_1, \beta_1, \gamma_1)\}$ to construct B1. Then, we flip B1 with respect to the reflective plane to generate B2 by $\{P_2: (x_1, y_1, -z_1)\}$, $\{\theta_2: (\alpha_1, \beta_1, -\gamma_1)\}$. Since the object is symmetric, the DDF values of corresponding rays in B1 and B2 should be the same, which establishes our symmetry loss term. (L. 206 in the manuscript). In the final version, we refine our explanation in L. 207—L.209 by adding mathematical symbols and also a figure to illustrate the construction of the symmetry loss in the Supplementary Material.
>
> ## Q3. Minor Comments
> Thank you very much for the detailed suggestions.
>
> (1) We'll consult with native speakers to identify a more fitting term to characterize the SDF's nature. We acknowledge that 'undirected' primarily pertains to graph descriptions, and while we are yet to discover a superior alternative, we remain open to suggestions.
>
> (2) We checked the typos again and corrected them.
>
> (3) Arbitrarily —-> Randomly
>
> (4) The positions of the figures are adjusted in the final version.
>
> (5) All citations are checked again.
>
> (6) Thank you again for the suggestion. We also find the information introduced in Sec. 3 is very important to understand our method. We put it in the main paper in the final verison.

---

> > ### Comment · Reviewer_hGDA · 2023-08-16
> > **Comments on rebuttal**
> >
> > I appreciate the authors’ efforts in responding to all comments from a large number of reviewers. I have read all reviews and responses, but I will limit this post to my comments. I will only point out that the additional ablation studies are useful.
> >
> > The response to W1 is informative. The proposed architecture is not much heavier than IHOI, but inference is about 4 times slower. A summary of the detailed response should be included in the paper.
> >
> > The response to W2 reveals lack of clarity in the paper. In light of the response, lines 38-40 over-emphasize the shortcomings of previous work when handling symmetric objects. Later on the same page, the geometric loss that handles symmetry is presented as an important contribution of the paper. Lines 243-245 indeed state that the symmetry loss is not included in most experiments for fairness. In my opinion, some rewriting is warranted to present symmetry more clearly. This is not an argument for rejecting the paper by any stretch.

---

> > > ### Author Response · Authors · 2023-08-17
> > > **Response to Reviewer hGDA**
> > >
> > > Thank you again for taking the time to thoroughly read our paper! We are very pleased that you can acknowledge our efforts. Thanks.
> > >
> > > After carefully reading your comments, we have also realized that our description of symmetry loss was not clear enough. In the final version, we reorganized the logic concerning the introduction of symmetry loss and provided more details. We have employed the following intuitive explanation in the 'introduction' section to clarify why SDF cannot naturally model the symmetry of objects.
> > >
> > > ### SDF has inherent theoretical flaws in modeling object symmetry.
> > >
> > > @@@@@reconstructed surface@@@@@
> > >
> > > *P
> > >
> > > \
> > >
> > > ~~reflective plane~~
> > >
> > > @@@@@reconstructed surface@@@@@
> > >
> > > *Q
> > >
> > > The above table can be viewed as a 2D extreme case illustrating why SDF fails to model symmetry.
> > > The reconstructed surface is marked with '@'.
> > > The ground truth object surface should be symmetric with respect to the reflective plane.
> > > Obviously, the current reconstruction is wrong and not symmetric.
> > > Therefore, using SDF as shape representation, we can also sample two bijection sets (but only sample points not directions) to enforce a symmetry loss like in DDF-HO.
> > > Take the two randomly sampled point-pair P and Q as an example.
> > > It can be clearly seen from the above table that although the reconstruction is wrong, P and Q still have the same SDF value!
> > > If this happens in training, it will confuse the network and further impair the performance.
> > >
> > > In summary, although such extreme cases are rare in real-world experiments, it still verifies that SDF has inherent theoretical flaws in modeling object symmetry.
> > > Therefore, SDF can not 'naturally' model object symmetry, unless some additional check is adopted to remove such cases.
> > > DDF can solve this problem by incorporating directions into the input.
> > >
> > > We added this intuitive explanation in the final version and hope this can help readers quickly catch the superiority of using DDF in hand-held object reconstruction. Thanks again.

---

### Official Review · Reviewer_dmTA · 2023-07-04

**Soundness:** 3 good
**Presentation:** 2 fair
**Contribution:** 3 good
**Rating:** 6
**Confidence:** 5

**Summary:**

This work proposes a novel pipeline that uses DDF as the shape representation for hand-held object reconstruction from a single image. DDFs provide benefits over SDFs, eg. they are directed, provide intersection with object information, and can capture symmetry. Extensive experiments on ObMan, HO3D, and MOW datasets show the effectiveness of the proposed approach over existing methods.

**Strengths:**

- This work proposes to use DDF, which is more expressive than SDF, for hand-held object reconstruction. While SDFs are undirected and cannot capture symmetry, DDFs provide a directed distance to the surface along with intersection with object information (visibility) and can capture symmetry.
- The proposed approach uses ray-based feature aggregation and intersection-aware hand features to better capture hand-object interaction compared to existing SDF-based methods.
- Extensive experiments on ObMan (Table 3), HO3D (Table 1) and MOW (Table 2) datasets show the effectiveness of the proposed approach over existing methods.
- Ablation studies on different components (Table 4) and robustness to noise in hand pose (Table 5) are helpful in understanding the capabilities of the proposed approach. Also, error bars over 5 training seeds are provided in the supplementary.

**Weaknesses:**

- The HO3D splits used are different than IHOI[62]. Is there any reason for this?
- The IHOI[62] scores on MOW (Table 2) are different than those reported in IHOI paper, even though the same splits are used (L221-222). Why is this the case?
- It'd be useful to have ablations on the different ray sampling strategies used during training, as stated in Sec. 3 in the supplementary. Specifically, how well does uniform sampling perform by itself? This could be a limitation when scaling DDF-HO to cases where 3D ground truth object models are not available (eg. in-the-wild settings).
- How does the training & inference time for DDF-HO compare to IHOI? Since several rays need to be sampled, it seems that DDF-HO would be slower than IHOI.
- It'd be helpful to include more details about ray sampling during testing in the main paper, eg. how many rays, how are the origin & directions sampled.

**Questions:**

Clarifications required are mentioned in the weaknesses above.

**Limitations:**

Limitations are discussed.

---
I appreciate the additional ablations and clarifications provided by the authors. After reading the rebuttal, other reviews, and discussion, I think that the authors have addressed the main concerns pointed out in the reviews. So, I am retaining my rating of `Weak Accept`.

---

> ### Author Rebuttal · Authors · 2023-08-08
>
> Thank you very much for taking the time to thoroughly review our paper. Your insights are highly valuable and provide us with guidance to enhance the paper. Below we try our best to address your concerns and questions. We have revised the paper according to your suggestions.
>
> ## Q1. Different Splits with IHOI
> The released code of IHOI is incomplete. We found that the preprocessed SDF data provided by IHOI doesn’t match its split files on HO3D and MOW. Thus we contacted the first author of IHOI for this issue in early April, 2023 during development of DDF-HO. She replied that her released data is not the same as used in the experiments in the paper. Under her help, we re-trained and re-evaluated all baseline methods on the two datasets using the split files provided in the released code of IHOI. Therefore, the results on HO3D and MOW are a little different from the results reported in IHOI.
>
> ## Q2. Different Results on MOW
> As explained in L. 221–L. 222, we follow the splits of the released code of IHOI. The split files in the released code are a little different from the ones used in the original paper. We use the new split files to evaluate the performance, leading to slightly different results.
>
> ## Q3. Ablations on Different Ray Sampling Strategies
> Thanks for the suggestion! We have conducted the ablations of different ray sampling methods on ObMan in the following table (Sampling methods 1 -- 5 are introduced in Sec. 3 of the Supplementary Material). It can be seen clearly that applying multiple kinds of sampling strategies contributes to improvements in performance, which is also verified in [DDF]. Please note that in testing stage, DDF-HO only uses uniform sampling since no prior shape information is accessible (introduced in the first paragraph in Sec. 3 of the Supplementary Material). Therefore, it can handle in-the-wild hand-held objects.
>
> | |     F5|F10 |   CD|
> |------|------|------|------|
> 1                 |     0.53 | 0.66 | 0.17
> 1,2               |     0.53 | 0.66 | 0.16
> 1,2,3            |      0.54 | 0.66 | 0.15
> 1,2,3,4           |     0.55 | 0.66 | 0.15
> 1,2,3,4,5        |      0.55 | 0.67 | 0.14
>
>
> ## Q4. Efficiency of DDF-HO
> IHOI is a typical SDF-based hand-held object reconstruction method. We list detailed comparisons with IHOI in the following table.
>
> ||DDF|IHOI
> |------|------|------|
> Training time| 2d 8h |IHOI: 2d|
> Parameters|24.5M| 24.0M
> Running Speed|23ms/img (44FPS) |5.8ms/img (172FPS)|
>
> From the above table, it can be seen that our method runs slower than IHOI but still achieves real-time performance. All experiments in the above table are conducted on a single NVIDIA A100 GPU. The slower inference comes from the attention calculation for 2D ray-based feature aggregation and the more elabored 3D feature generation. Ray sampling in the testing time only takes less than 1ms since only randomly sampling on unit sphere for ray starting points and directions is conducted. For model size, the two methods share a similar scale. Generally, the increased parameters mainly come from the cross attention mechanism in 2D ray-based feature aggregation (Sec. 3.3). Other modules are only adopted to collect features to model hand-object interactions and do not significantly increase the network size. Moreover, since the network needs to predict both the distance and visibility signal, the convergence speed of DDF-HO during training is also slower than IHOI.
>
> ## Q5. Ray Sampling During Inference
> As introduced in L. 24—L. 25 in the supplementary material, during inference, the ground truth object in the datasets is already canonicalized and resized to fit in a unit sphere. Thus, we simply use the uniform sampling to generate 20K 3D rays. The origins are randomly sampled inside the sphere and the directions are also randomly sampled from the uniform distribution. Since the sampled 3D rays may not intersect with the object, the yielded point cloud from DDF prediction contains less than 20K points. Thus, we randomly sample 12K points for evaluation. For fair comparisons, all other methods are trained with 20K points and tested using 12K points (This is also in accordance with the released code of IHOI). We clarified this information in the final version.
>
> [DDF] Aumentado-Armstrong, T., Tsogkas, S., Dickinson, S., & Jepson, A. D. (2022). Representing 3D shapes with probabilistic directed distance fields. In Proceedings of the IEEE/CVF Conference on Computer Vision and Pattern Recognition (pp. 19343-19354).

---

> > ### Comment · Reviewer_dmTA · 2023-08-14
> > **Response to rebuttal**
> >
> > I have read the rebuttal and other views. I appreciate the additional ablations and clarifications provided by the authors.
> >
> > I have a few more questions:
> > - Which evaluation setting is used for ablation on ray sampling strategies (is it ObMan)? The difference seems to be marginal between different strategies. It'd be helpful to provide some insights into why this is the case. For training SDF-based models, points need to be sampled near the surface otherwise the model does not work well. In case of DDF, it seems like sampling rays uniformly is nearly as good as sampling near the surface.
> > - At inference, the model does not require any prior shape information. However, 3D ground truth is required at training time. Any thoughts on how this can be extended to setting where 3D ground truth is not available during training?

---

> > > ### Author Response · Authors · 2023-08-15
> > > **Response to Reviewer dmTA's Questions**
> > >
> > > We sincerely appreciate your valuable and constructive comments. Thank you again for taking the time to thoroughly review our paper!
> > >
> > > ## Performance Using Different Ray Sampling Strategies
> > > In Q3, we conduct experiments on ObMan. We keep the number of rays from each sampling strategy unchanged except the ablated one to demonstrate the overall effect of each sampling strategy in the reconstruction. As for the marginal performance gains when incrementally incorporating different sampling strategies, we identify the following possible reasons.
> > >
> > > First, among all the sampled 3D rays, those from uniform sampling make up half (10K rays), while the rest of the sampling strategies together make up the other half, as explained in detail in L.37 in Sec. 3 of the supplementary material. The dominance of uniform sampling is underscored by its usage in inference, where only uniform sampling is available due to lack of object priors. From the ablation study, only using rays from uniform sampling is sufficient to train a satisfactory model, but other sampling strategies still contribute a little improvement.
> > >
> > > Second, we follow the recommendation of DDF [2] to utilize all the 5 sampling strategies. "As the single field experiment shows (Sec. 4 in DDF [2] and Sec. C in the supplementary material of DDF [2]), each sampling method can collect information that the other types cannot completely make up for." In DDF [2], complex objects like sculptures of dragons and rabbits are used in inference, which highlights the role of different sampling strategies. In our paper, on ObMan, HO3D and MOW, the objects are hand-held and usually not very complex. Thus the effect of incorporating different sampling strategies is weakened. However, considering applying our method in real-world scenarios, complex hand-held objects are not rare. Using all 5 sampling methods can ensure stable and consistent performance.
> > >
> > > Third, reconstructing hand-held object is inherently very challenging. As other reviewers mentioned, the visualization results of DDF-HO are still not satisfactory, although we already surpass all baselines. Currently, the recovery of basic object shape is still not very accurate due to heavy occlusions from hand and lack of priors. Therefore, representing fine-grained higher-order geometry with different sampling strategies is not the main problem.
> > >
> > > Last but most importantly, our approach of ray-based feature aggregation mitigates the necessity for samples to be in close proximity to the object surface. By collecting features along rays, we effectively capture information from both the ray's starting and ending points, as comprehensively detailed in L.48-L.50 and Sec. 3.3 of the main paper. Our method allows good performance with only uniform sampling.
> > >
> > > ## Training Without 3D Ground Truth
> > > Thanks for the constructive comments. DDF-HO does not require any prior shape information at inference, which enables it to handle in-the-wild hand-held objects. Currently, all baseline methods need 3D ground truth for training, under the single-image reconstruction setting.
> > >
> > > Training without 3D ground truth is an interesting but challenging and open problem to be solved. We propose several tentative ideas here.
> > >
> > > First, reconstruction with multi-view input. Given multiple views describing the same hand-held object, we can establish epipolar geometry for supervision (like in [R-1]) and leverage the MANO hand model to provide coarse prior knowledge of the object shape.
> > >
> > > Second, reconstruction with single RGB-D image. This can be achieved via recent proposed SinNeRF [R-2]. However, how to deal with the occlusion caused by hand and leverage the hand pose information are still open problems.
> > >
> > > Third, reconstruction with single RGB image. Recently, a single image decomposition method [R-3] seems to be feasible in this case. Given a single image with known object category, [R-3] can directly recover the geometry and texture information after unsupervised training. However, such methods usually perform poorly in real-world scenes. Moreover, the occlusion caused by hand also significantly influences the image decomposition process.
> > >
> > > [R-1] Truong, Prune, et al. Sparf: Neural radiance fields from sparse and noisy poses. CVPR 2023.
> > >
> > > [R-2] Xu, Dejia, et al. SinneRF: Training neural radiance fields on complex scenes from a single image. ECCV 2022.
> > >
> > > [R-3] Monnier, Tom, et al. Share with thy neighbors: Single-view reconstruction by cross-instance consistency. ECCV 2022.

---

### Official Review · Reviewer_1CC4 · 2023-07-06

**Soundness:** 3 good
**Presentation:** 2 fair
**Contribution:** 3 good
**Rating:** 4
**Confidence:** 4

**Summary:**

This paper proposes a directed distance field-based method for hand-held object reconstruction from a single RGB image. The paper aggregates 2D ray-based features to capture ray-object intersection and 3D geometric features of ray-hand intersection. In particular, it extracts local to global cues via the above features and introduces a symmetry loss term to handle symmetric objects. Experiments on three datasets and ablations on the introduced modules show the effectiveness of the proposed method.

**Strengths:**

1. The proposed idea of extracting local 2D image features along the ray and local 3D features from the ray-hand relationship to provide geometric cues is novel in hand-held object reconstruction.

2. Supporting experiments show that DDF is a more suitable representation than SDF for handheld object reconstruction.

**Weaknesses:**

1. Unclear descriptions. It's better to provide a clearer description of how to sample the bijection sets on the reflective plane for symmetry loss, preferably with an illustration.

2. Insufficient qualitative results. It's suggested to provide qualitative results of the ablation studies, especially the ablation on symmetry loss term. Besides, it should be declared that whether adding such a symmetry loss term leads to a wrong distance field for asymmetric objects.

3. Insufficient ablations on input hand poses. It would be better to add Gaussian noises at a larger scale (e.g., \sigma=1.0,1.5) to the input hand poses.

4. Insufficient ablations on K_l and K_{3D}. Why K_l=8 is set to sample points along the projected 2D ray. And why K_{3D}=8 (nearly half of hand joints) is set to select the neighboring hand joints, since considering fewer hand joints seems to be more efficient for capturing local ray-hand features.

**Questions:**

1. The baselines mentioned in the paper (HO, GF, AlignSDF, and IHOI) all consider the intersection between hand and object and report the corresponding metric Intersection Volume [1], which are not mentioned in this paper. The authors should provide experimental results of Intersection Volume and state why the proposed method is able to outperform other methods on this metric (if applicable) without considering the contact/intersection between hands and objects.

2. In this paper, the canonical hand shape of MANO is used and only the articulation parameters of the hand are taken for modeling. However, hand shape is also a geometric cue, important for hand-held object reconstruction, but difficult to learn, which is considered in the baselines (HO and GF). The authors should make sure that they are comparing these baselines in a fair setting.

[1] Hasson, Y., Varol, G., Tzionas, D., Kalevatykh, I., Black, M.J., Laptev, I., Schmid, C.: Learning joint reconstruction of hands and manipulated objects. In: CVPR (2019)

**Limitations:**

The paper has well clarified the limitations of the proposed methods, which are inherited from the shortcomings of DDF.

---

> ### Author Rebuttal · Authors · 2023-08-09
>
> We sincerely appreciate your valuable and constructive comments. Our detailed responses are listed below and we revised the manuscript accordingly.
>
> ## Q1. Unclear Descriptions of Symmetry Loss
> Following other baseline methods, we exclude the symmetry loss except for ablation studies (L. 243 in the manuscript). Hence, our results across different datasets and visualizations are symmetry loss-free. Tab. 4 shows a slight improvement due to the symmetry loss. Our updated paper and the uploaded PDF attachment now include qualitative object comparisons with and without the symmetry loss.
>
> Before our experiments, all symmetric objects in the datasets are preprocessed to be symmetric with respect to the XY plane $\{x=0, y=0\}$ . To determine whether the object is symmetric, we first flipping the sampled points P on the object surface w.r.t the XY plane, yielding P'. Then we compare the Chamfer Distance between the object surface and P'. If the distance lies below a threshold (1e-3), the object is considered symmetric. As for building two bijection sets B1 $\{P_1, \theta_1\}$ and B2 $\{P_2, \theta_2\}$, we first randomly sample origins $\{P_1: (x_1, y_1, z_1)\}$ and directions $\{\theta_1: (\alpha_1, \beta_1, \gamma_1)\}$ to construct B1. Then, we flip B1 with respect to the reflective plane to generate B2 by $\{P_2: (x_1, y_1, -z_1)\}$, $\{\theta_2: (\alpha_1, \beta_1, -\gamma_1)\}$. Since the object is symmetric, the DDF values of corresponding rays in B1 and B2 should be the same, which establishes our symmetry loss term. (L. 206 in the manuscript). In the final version, we refine our explanation in L. 207—L.209 by adding mathematical symbols and also a figure (shown in the uploaded attachment) to illustrate the construction of the symmetry loss in the Supplementary Material.
>
>
> ## Q2. Insufficient Ablations on Symmetry Loss
> Please note that adding this symmetry loss is a natural choice of using DDF representation. Therefore, symmetry loss is not our main contribution. For fair comparisons, we do not use the symmetry loss except in the ablation studies in Tab. 4, as explained in L.243 in the manuscript. Moreover, as explained in L. 207, we apply this loss only on symmetric objects during training. Thus, it should make little influence on the asymmetric objects. Such observation is made on our ablations shown in the table below.
> ||F5 |  F10 |   CD
> |-|-|-|-|
> Symmetric objects (w/o symmetry loss) |0.57 | 0.69  |0.12|
> Asymmetric objects (w/o symmetry loss)|0.50 | 0.62|0.17|
> Symmetric objects (w/ symmetry loss) | 0.59 | 0.70 | 0.11|
> Asymmetric objects (w/ symmetry loss)| 0.51 | 0.62|0.16|
>
> ## Q3. Additional Experiments on the Noise of Hand Pose
> We also conduct hand pose ablations with larger Gaussian noise on ObMan (the upper block) and HO3D (the bottom block). The results are exhibited below. Adding too large noise on hand poses will lead to drop on the reconstruction precision (F5). However, the CD metric remains to be 0.31 on ObMan even with sigma=1.5 Gaussian noise, which is still lower than SDF based baselines. This demonstrates the robustness of DDF.
>
> ||F5 |  F10 |   CD
> |-|-|-|-|
> Pred (no noise)        |            0.55 | 0.67 | 0.14|
> sigma=1.0 |              0.42  |0.55 | 0.25|
> sigma=1.5  |             0.38 | 0.52 | 0.31|
> Pred  (no noise)  |     0.28  |0.42  |0.55|
> sigma=1.0|0.17 | 0.27  |0.98|
> sigma=1.5   |            0.14|0.25 | 1.24|
>
> ## Q4. Additional Ablations on $K_l$ and $K_{3D}$
> We show the relevant ablations of $K_l$ and $K_{3D}$ on ObMan in the table below. When setting $K_l=12$, the performance will be enhanced a little, but with the cost of larger network parameters and slower inference speed. Increasing $K_{3D}$ to $12$ does not provide any improvements towards the reconstruction since (1) DDF-HO has already considered the global hand feature $F_{3D}^G$; (2) collecting local intersection feature $\mathcal{F}_{3D}^L$ in a larger field will not provide more local information.
>
> |keep $K_{3D}=8$   |     F5|F10 |   CD|
> |-|-|-|-|
> $K_l=4$        |       0.53 | 0.65 | 0.16|
> $K_l=8$        |       0.55 | 0.67 | 0.14|
> $K_l=12$        |      0.56 | 0.68 | 0.13|
> |keep $K_{l}=8$ |||
> $K_{3D}=4$       |     0.54 | 0.66 | 0.15|
> $K_{3D}=8$|0.55|0.67  |0.14|
> $K_{3D}=12$     |      0.55 | 0.67|  0.14|
>
> ## Q5. Results w.r.t Intersection Volume
> Results are shown in the following table. It can be seen clearly that DDF-HO outperforms IHOI by 0.07 on ObMan and 0.57 on HO3D. Our method explicitly models the hand-object interactions, or in other words, 'considers the contact/intersection between hands and objects', in two aspects. First, on the 2D image, hand information along the projected 2D ray is encoded into the corresponding 2D features $F_{2D}$ (Sec. 3.3). Second, Our 3D embeddings $F_{3D}$ encapsulate both global hand shape feature and local hand-object geometric features (Sec.3.4).
>
> ||   ObMan|HO3D |
> |-|-|-|
> |HO |8.64|                     10.03 |
> |GF |1.84 |                    7.16|
> |IHOI| 1.74|                  4.92|
> |Ours| 1.67 |                 4.35 |
>
> ## Q6. Fair Comparison with Baselines
> We strictly follow the setting of the SDF-based baseline IHOI. Given an RGB image and predicted hand pose as input, we reconstruct the hand-held object. Therefore, our comparisons with IHOI are strictly fair. For other methods, since they also predict the hand shape while we directly get the hand mesh with the predicted hand pose and MANO models, we mainly compare the object reconstruction quality with them, like IHOI in its paper. Sharing the exactly same setting towards IHOI, we can ensure the fairness when comparing with such methods. On an overall level, our method and all other methods can start from an RGB image and ultimately obtain the reconstructed result of the hand-held object. Therefore the comparisons are supposed to be fair.

---

### Official Review · Reviewer_e2Ui · 2023-07-07

**Soundness:** 3 good
**Presentation:** 3 good
**Contribution:** 3 good
**Rating:** 8
**Confidence:** 2

**Summary:**

The authors introduce a novel approach called DDF-HO, which uses Directed Distance Field (DDF) for 3D hand-held object reconstruction. Unlike SDF, DDF includes origins and directions of the views in the 3D space. They show that the ray-based feature aggregation scheme and 3D intersection-aware hand pose embeddings are more suitable for 3D hand-held object reconstruction. The DDF-HO method outperforms prior work.

**Strengths:**

1. Proposed a new data structure DDF for hand-held object reconstruction.
2. Ray-Based Feature Aggregation technique has better representations of local geometric features.
3. The interaction modeling reflects the interaction between the hand and object rather than just hand as in the prior work.
4. Significant improvements over prior work.
5. Comprehensive evaluation and ablation studies.

**Weaknesses:**

1. Lack of efficiency comparison between SDF-based methods and proposed DDF. Such as the amount of resources required for reconstructing the same input.
2. No appearance on the reconstruction results

**Questions:**

1. How long does it take to reconstruct one image?

**Limitations:**

1, As the authors mentioned, it is hard to train DDF on higher dimensional inputs.
2. As the authors mentioned, the reconstructions do not reflect the translucency, material, and appearance of the objects
3. The reconstruction quality is not good, especially on real-world images

---

> ### Author Rebuttal · Authors · 2023-08-08
>
> We sincerely appreciate you for the precious review time and valuable comments. We revised the manuscript according to the review. Below we try our best to address your concerns and questions.
>
> ## Q1. Efficiency
> IHOI is a typical SDF-based hand-held object reconstruction method. We list detailed comparisons with IHOI in the following table.
>
> ||DDF|IHOI
> |------|------|------|
> Training time| 2d 8h |IHOI: 2d|
> Parameters|24.5M| 24.0M
> Running Speed|23ms/img (44FPS) |5.8ms/img (172FPS)|
>
> From the above table, it can be seen that our method runs slower than IHOI but still achieves real-time performance.All experiments in the above table are conducted on a single NVIDIA A100 GPU. For model size, the two methods share a similar scale. Generally, the increased parameters mainly come from the cross attention mechanism in 2D ray-based feature aggregation (Sec. 3.3). Other modules are only adopted to collect features to model hand-object interactions and do not significantly increase the network size. Moreover, since the network needs to predict both the distance and visibility signal, the convergence speed of DDF-HO during training is also slower than IHOI.
>
> ## Q2. No Appearance in the Reconstruction Results
> In this paper, we focus on recovering the geometric shape of the hand-held object. As other reviewers point out, currently reconstructing the shape of hand-held object without prior knowledge of the object is still a challenging and open problem, since no existing methods demonstrate satisfactory performance. Experiments showcase that our method achieves state-of-the-art performance in real-world and synthetic datasets. As for the appearance, in the future work, a coloring network like in PC2 [PC2] can be adopted to predict the color for each pixel. Moreover, since DDF shares similar input as NeRF, neural view synthesis may be also possible by slightly adjusting the DDF representation, which will make the problem very interesting.
>
> [PC2] Melas-Kyriazi, L., Rupprecht, C., \& Vedaldi, A. (2023). PC2: Projection-Conditioned Point Cloud Diffusion for Single-Image 3D Reconstruction. In Proceedings of the IEEE/CVF Conference on Computer Vision and Pattern Recognition (pp. 12923-12932).

---

> > ### Comment · Reviewer_e2Ui · 2023-08-21
> >
> > Thank you for providing the efficiency experiments and addressing my questions.
> >
> > I think all my concerns have been satisfactorily addressed, and my rating continues as Strong Accept, as the proposed Ray-Based Feature Aggregation method opens new ways for hand-held object reconstruction, and compared to previous SDF-based methods, the new method has better reconstruction results and as stated still achieves real-time performance.

---

### Official Review · Reviewer_7sso · 2023-07-08

**Soundness:** 3 good
**Presentation:** 3 good
**Contribution:** 3 good
**Rating:** 4
**Confidence:** 3

**Summary:**

The paper proposes an algorithm for reconstructing hand-held object from a single RGB image. Instead of using the traditional Signed Distance Fields (SDF), this paper proposes to leverage Directed Distance Field (DDF) as the shape representation. Experiment shows that the proposed algorithm outperforms SOTA.

**Strengths:**

The proposed pipeline utilizes DDF as the shape representation to reconstruct hand-held objects, which has a stronger modelling capability for this specific task, e.g., reconstruction for hand-held objects from a single RGB image. It introduce a 2D ray-base feature aggregation and 3D intersection-aware hand pose embedding.
The experiments are conducted on both real and synthetic dataset,

**Weaknesses:**

The paper did not mention the running speed, model complexity, I wonder if it's comparable with SDF based models.
The paper mentioned that the DDF is harder to train, and requires more complex data, algorithm, and network structure, I wonder if the paper can give more quantitative measurement? e.g. 1 or 2 magnitude harder/longer/more parameters?

Another alternative to SDF will be the Occupancy, the paper did not mention Occupancy at all. Won't Occupancy be a strong baseline model? or replacing SDF with Occupancy will make the algorithm (proposed and SOTA) perform better?

**Questions:**

see weaknesses

**Limitations:**

did not talk about potential negative societal impact

---

> ### Author Rebuttal · Authors · 2023-08-08
>
> Thank you for the constructive review. We revised the manuscript accordingly. Below we try our best to address your concerns and questions.
>
> ## Q1. Running Speed and Model Complexity
> IHOI is a typical SDF-based hand-held object reconstruction method. We list detailed comparisons with IHOI in the following table.
>
> ||DDF|IHOI
> |------|------|------|
> Training time| 2d 8h |IHOI: 2d|
> Parameters|24.5M| 24.0M
> Running Speed|23ms/img (44FPS) |5.8ms/img (172FPS)|
>
> From the above table, it can be seen that our method runs slower than IHOI but still achieves real-time performance.All experiments in the above table are conducted on a single NVIDIA A100 GPU. For model size, the two methods share a similar scale. Generally, the increased parameters mainly come from the cross attention mechanism in 2D ray-based feature aggregation (Sec. 3.3). Other modules are only adopted to collect features to model hand-object interactions and do not significantly increase the network size. Moreover, since the network needs to predict both the distance and visibility signal, the convergence speed of DDF-HO during training is also slower than IHOI.
>
> ## Q2. Occupancy as the Shape Representation
> Since no off-the-shelf hand-held object reconstruction pipelines leverage occupancy as the shape representation, we design a baseline method ourselves following the widely-used 2D-3D lifting schemed single-view reconstruction method [Pix2Vox]. We first use the same backbone as our method (ResNet34) to extract per-pixel features. The extracted features are then back-projected to the volume (32x32x32, the same as [Pix2Vox]). For each voxel inside the volume, hand pose is also concatenated to its feature vector. Finally, we predict the occupancy as in Pix2Vox [Pix2Vox]. The results on ObMan (the upper block) and HO3D(V2) (the bottom block) are shown as follows.
>
> |Method|F5|F10|CD|
> |------|------|------|------|
> |IHOI| 0.42 | 0.63 | 1.02  |
> |DDF-HO| 0.55 |0.67 |0.14  |
> |Pix2Vox |0.24 |0.45 |1.81 |
> |------|------|------|------|
> |IHOI|0.21 |0.38 |1.99|
> |DDF-HO|0.28 |0.42 |0.55|
> |Pix2Vox |0.06 |0.17 |6.12|
>
> Thereby we explain why occupancy is also not a suitable choice in hand-held object reconstruction. First, like SDF, occupancy representation is also undirected. It can only model the local shape of either object or hand. For voxels that are not near the hand, the occupancy representation also cannot naturally capture the relationship between the hand and the object. Second, the reconstruction quality is limited by the resolution of the volume.  Since the entire volume is typically uniformly divided into voxels, for object regions with particularly complex geometric structures, large voxels might limit the network's expressive capacity, resulting in decreased accuracy of the results.
>
> [Pix2Vox] Xie, H., Yao, H., Sun, X., Zhou, S., \& Zhang, S. (2019). Pix2vox: Context-aware 3d reconstruction from single and multi-view images. In Proceedings of the IEEE/CVF international conference on computer vision (pp. 2690-2698).

---

### Official Review · Reviewer_9guR · 2023-07-27

**Soundness:** 2 fair
**Presentation:** 1 poor
**Contribution:** 2 fair
**Rating:** 3
**Confidence:** 3

**Summary:**

This paper presents a system for joint hand and hand-held object 3D reconstruction. The authors propose a pipeline to (1) predict the hand (MANO hand model) and camera poses with an off-the-shelf pose estimator; (2) extract image features with an off-the-shelf ResNet; (3) sample "3D ray representations", project them to the 2D image plane, and extract the corresponding image features and aggregating them with cross-attention along the ray directions; (4) for the same 3D points, extract 3D features from the global hand embedding from the geodesic nearest joint; (5) predict a directed distance function (DDF) with an MLP, using the above features as input. Experiments show that the proposed system predicts more accurate shapes.

**Strengths:**

- The proposed method is evaluated on synthetic and real hand-object interaction datasets, showing improvements over baselines on nearly all metrics. Ablative analyses are also provided to better understand the behaviors of the proposed DDF-HO.

**Weaknesses:**

- The presentation is poor. Specifically, it is unclear what the use of ray sampling is, and how the random ray directions transform to the directions pointing to the target hand (R-A to R-B in Fig 2). Also, it is unclear how one would know a sampled ray would (not) point to the direction closest to the hand shape, without the DDF even being optimized. In addition, it seems that the major contribution of the paper is a system proposal as a whole, rather than the choice of representation (DDF vs. SDF). I think the message of the paper is somewhat misleading in this sense.
- Why is the final output in the form of DDF, if the end goal is to extract the surface of the hand shape? Predicting an SDF or occupancy field would serve the same purpose as well. Also, it is not clear how DDF is being taken advantage of in the proposed system. The evaluation metric is based on surface point cloud representations as well.
- It is not clear to me how the proposed method is better in the real-world datasets. Visually, they seem to be very far from good predictions of the object shapes.
- It would be good to analyze failure cases to better understand the limitations of the proposed system.

**Questions:**

It would be great if the authors could address the questions raised in the weakness section.

**Limitations:**

Yes

---

> ### Author Rebuttal · Authors · 2023-08-08
>
> We sincerely appreciate your valuable and constructive comments. We are delighted that you found our method “predicts more accurate shape” than competitors. Our detailed responses are listed below and we revise the manuscript accordingly.
>
> ## Q1. About Ray Sampling from (R-A) to (R-B)
> We define DDF in L.106 — L.113 in Sec. 3.1. It maps a 3D ray (consisting of an origin and a view direction) to a non-negative scalar field (L.109) and a binary visibility field (L.111). The binary visibility field plays a crucial role in identifying which sampled rays do not intersect with the hand-held object. This identification is significant as these non-intersecting rays do not contribute to the overall reconstruction process. Therefore, in the pipeline Fig. 2, we only showcase the rays that are necessary in reconstruction. The ray sampling algorithm is introduced in detail in Sec. 3 in the Supplementary Material. To avoid misleading the readers, we have incorporated a distinct color to denote rays that do not intersect with the object in Fig. 2 in the final version. This visual distinction serves to enhance the accuracy of the demonstration.
>
> ## Q2. Our Contributions
> We first explain why SDF is not a suitable choice for hand-held objection reconstruction in our paper (L.27 – L.40 in Introduction, L.102 – L.105 in Sec. 3.1, L.142 – L.159 in Sec. 3.3). SDF is undirected and too compact that fails to naturally encapsulate local interactions between hand and the object (L.27 – L.40). Then we point out that DDF is a better choice and design DDF-HO which utilizes DDF as the shape representation for hand-held object reconstruction. We list our contributions in L.60—L.66 in the main paper. We briefly summarize them here for clarity. We present DDF-HO, a novel pipeline that utilizes DDF for hand-held reconstruction, outperforming SDF-based competitors. Based on the DDF representation, we propose a novel ray-based feature aggregation scheme to model hand-object relationship, which boosts the overall reconstruction quality. Extensive experiments and ablation studies demonstrate the effectiveness of DDF-HO.
>
> ## Q3. Occupancy as the Shape Representation
> Since no off-the-shelf hand-held object reconstruction pipelines leverage occupancy as the shape representation, we design a baseline method ourselves following the widely-used 2D-3D lifting scheme in single-view reconstruction [Pix2Vox]. We first use the same backbone as our method (ResNet34) to extract per-pixel features. The extracted features are then back-projected to the volume (32x32x32, the same as [Pix2Vox]). For each voxel inside the volume, the predicted hand pose (the same as used in DDF-HO, parameterized as the MANO model parameters) is also concatenated to its feature vector. Finally, we predict the occupancy as in Pix2Vox [Pix2Vox]. The results on ObMan (the upper block) and HO3D(V2) (the bottom block) are shown as follows.
>
> |Method|F5|F10|CD|
> |------|------|------|------|
> |IHOI| 0.42 | 0.63 | 1.02  |
> |DDF-HO| 0.55 |0.67 |0.14  |
> |Pix2Vox |0.24 |0.45 |1.81 |
> |------|------|------|------|
> |IHOI|0.21 |0.38 |1.99|
> |DDF-HO|0.28 |0.42 |0.55|
> |Pix2Vox |0.06 |0.17 |6.12|
>
>
> Thereby we explain why occupancy is also not a suitable choice in hand-held object reconstruction. First, like SDF, occupancy representation is also undirected. It can only model the local shape of either object or hand. For voxels that are not near the hand, the occupancy representation also cannot naturally capture the relationship between the hand and the object. Second, the reconstruction quality is limited by the resolution of the volume. Since the entire volume is typically uniformly divided into voxels, for object regions with particularly complex geometric structures, large voxels might limit the network's expressive capacity, resulting in decreased accuracy of the results.
>
> ## Q4. DDF in DDF-HO
> We use DDF to represent the 3D shape in our pipeline DDF-HO. As introduced in Sec. 3.5 in the paper, after extracting corresponding features of each ray (Sec. 3.3 and Sec.3.4), we leverage an 8-layer MLP to map the features to the corresponding DDF values: distance and binary visibility signal. (L.139 – L.140, L.200 – L.201). DDF can then be converted into other commonly used 3D representations including point cloud, mesh and vanilla SDF (L.112 – L.113), using the algorithms introduced in [2, 27].The superiority of DDF to SDF is accentuated in our main paper (L.27 - L.46, and also in Fig. 1). We demonstrate the DDF is a more suitable representation for hand-held object reconstruction.
>
> ## Q5. Unsatisfactory Visualization Results
> Our goal is to reconstruct the hand-held object from a single RGB image without any prior knowledge of the object. This is a recently emerging and very challenging research topic. Although our method still does not completely solve the problem, we make a significant leap forward in accuracy compared to previous methods (Tab. 1, 2, 3). In the future, combining other techniques, like diffusion models, with our DDF based reconstruction framework may further boost the performance.
>
> ## Q6. Failure Cases
> MOW pencil is a typical failure case (Row 3 in Fig. 3 of the Supplementary Material). Currently, reconstructing very thin objects is still a big challenge for all methods. In the final version and the submitted PDF attachment, we have added more visual results of the failure cases to comprehensively showcase our method.
>
> [Pix2Vox] Xie, H., Yao, H., Sun, X., Zhou, S., \& Zhang, S. (2019). Pix2vox: Context-aware 3d reconstruction from single and multi-view images. In Proceedings of the IEEE/CVF international conference on computer vision (pp. 2690-2698).

---

### Author Rebuttal · Authors · 2023-08-09

We sincerely appreciate the valuable work by all ACs and reviewers. We are delighted that DDF-HO is considered to show "improvements over baselines on nearly all metrics" [9guR, e2Ui, dmTA, hGDA], "stronger modelling capability" [7sso], "novelty in hand-held object reconstruction" [1CC4]. We revised the manuscript according to the suggestions. Below we try our best to address the common concerns and questions.


## Q1. Efficiency [All Reviewers]
IHOI is a typical SDF-based hand-held object reconstruction method. We list detailed comparisons with IHOI in the following table.

||DDF|IHOI
|------|------|------|
Training time| 2d 8h |IHOI: 2d|
Parameters|24.5M| 24.0M
Running Speed|23ms/img (44FPS) |5.8ms/img (172FPS)|

From the above table, it can be seen that our method runs slower than IHOI but still achieves real-time performance.All experiments in the above table are conducted on a single NVIDIA A100 GPU. For model size, the two methods share a similar scale. Generally, the increased parameters mainly come from the cross attention mechanism in 2D ray-based feature aggregation (Sec. 3.3). Other modules are only adopted to collect features to model hand-object interactions and do not significantly increase the network size. Moreover, since the network needs to predict both the distance and visibility signal, the convergence speed of DDF-HO during training is also slower than IHOI.

## Q2. Occupancy as the Shape Representation [9guR, 7sso]
Since no off-the-shelf hand-held object reconstruction pipelines leverage occupancy as the shape representation, we design a baseline method ourselves following the widely-used 2D-3D lifting scheme in single-view reconstruction [Pix2Vox]. We first use the same backbone as our method (ResNet34) to extract per-pixel features. The extracted features are then back-projected to the volume (32x32x32, the same as [Pix2Vox]). For each voxel inside the volume, the predicted hand pose (the same as used in DDF-HO, parameterized as the MANO model parameters) is also concatenated to its feature vector. Finally, we predict the occupancy as in Pix2Vox [Pix2Vox]. The results on ObMan (the upper block) and HO3D(V2) (the bottom block) are shown as follows.

|Method|F5|F10|CD|
|------|------|------|------|
|IHOI| 0.42 | 0.63 | 1.02  |
|DDF-HO| 0.55 |0.67 |0.14  |
|Pix2Vox |0.24 |0.45 |1.81 |
|------|------|------|------|
|IHOI|0.21 |0.38 |1.99|
|DDF-HO|0.28 |0.42 |0.55|
|Pix2Vox |0.06 |0.17 |6.12|


Thereby we explain why occupancy is also not a suitable choice in hand-held object reconstruction. First, like SDF, occupancy representation is also undirected. It can only model the local shape of either object or hand. For voxels that are not near the hand, the occupancy representation also cannot naturally capture the relationship between the hand and the object. Second, the reconstruction quality is limited by the resolution of the volume. Since the entire volume is typically uniformly divided into voxels, for object regions with particularly complex geometric structures, large voxels might limit the network's expressive capacity, resulting in decreased accuracy of the results.

## Q3. Symmetry [hGDA, 1CC4]
We provide a figure in the attachment to demonstrate the sampling method of building the bijection sets for the symmetry loss. We also provide qualitative comparisons of results with or without symmetry loss.

---

### Author Response · Authors · 2023-08-18
**Looking Forward to the Reviewers' Replies**

Dear reviewers,

Thank you for dedicating your time and attention to reviewing our paper. We have tried our best to address all the highlighted concerns and suggestions. We kindly request your consideration of our response. Your feedback holds immense value for us, and we are open to engaging in further discussions to enhance the quality of our work.

Best regards,

Paper 3312 Authors

---

### Decision · Program_Chairs · 2023-09-21

**Decision:**

Accept (poster)

**Comment:**

This submission received the following scores: R, BR, SA, BR, WA, WA

The authors complained about the "Reject" reviewer, and I agreed with them it seems the reviewer has a very superficial read of the paper.

There was some ethic concerns at some point (about skin color) but I think this aspect is solved - the datasets have various skin tones.

Overall, the reviewers are positive about the contribution and experiments. The authors gave additional, better explanations about the contribution and this new content should be included in the final version.